# Complete fusion of a transposon and herpesvirus created the *Teratorn* mobile element in medaka fish

Yusuke Inoue[1], Tomonori Saga[1], Takumi Aikawa[1], Masahiko Kumagai[1], Atsuko Shimada[1], Yasushi Kawaguchi[2], Kiyoshi Naruse [3], Shinichi Morishita[4], Akihiko Koga[5] & Hiroyuki Takeda[1]

Mobile genetic elements (e.g., transposable elements and viruses) display significant diversity with various life cycles, but how novel elements emerge remains obscure. Here, we report a giant (180-kb long) transposon, *Teratorn*, originally identified in the genome of medaka, *Oryzias latipes*. *Teratorn* belongs to the *piggyBac* superfamily and retains the transposition activity. Remarkably, *Teratorn* is largely derived from a herpesvirus of the *Alloherpesviridae* family that could infect fish and amphibians. Genomic survey of *Teratorn*-like elements reveals that some of them exist as a fused form between *piggyBac* transposon and herpesvirus genome in teleosts, implying the generality of transposon-herpesvirus fusion. We propose that *Teratorn* was created by a unique fusion of DNA transposon and herpesvirus, leading to life cycle shift. Our study supports the idea that recombination is the key event in generation of novel mobile genetic elements.

[1] Department of Biological Sciences, Graduate School of Science, The University of Tokyo, 7-3-1 Hongo, Bunkyo-ku, Tokyo 113-0033, Japan. [2] Division of Molecular Virology, Department of Microbiology and Immunology, The Institute of Medical Science, The University of Tokyo, 4-6-1 Shirokanedai, Minato-ku, Tokyo 108-8639, Japan. [3] Laboratory of Bioresources, National Institute for Basic Biology, 38 Nishigonaka, Myodaiji, Okazaki, Aichi 444-8585, Japan. [4] Department of Computational Biology and Medical Sciences, Graduate School of Frontier Sciences, The University of Tokyo, 5-1-5 Kashiwanoha, Kashiwa, Chiba 277-8561, Japan. [5] Primate Research Institute, Kyoto University, 41-2 Miyabayashi, Inuyama, Aichi 484-8506, Japan. Correspondence and requests for materials should be addressed to H.T. (email: htakeda@bs.s.u-tokyo.ac.jp)

Mobile genetic elements are genetic entities that can move inside genomes of cellular organisms or between cells. These elements include transposable elements (TEs) and viruses as well as plasmids and self-splicing elements in bacteria[1]. All elements share the ability to parasitize cells and propagate by usurping the host machinery. In addition, all elements contain "hallmark genes" associated with replication (e.g., polymerase, helicase, transposase) and structural constitution (e.g., capsid protein), which are not shared by cellular organisms[2]. Despite those common characteristics, each class of elements has a different genome constitution (either DNA or RNA of double-stranded (ds) or single-stranded (ss) nucleotides), replication manner (e.g., DNA polymerization, RNA-dependent RNA transcription, reverse transcription) and life cycle (i.e., TEs for persistent stay and propagation inside host genomes and viruses for transient stay and massive transfer between cells). How does this diversity arise and what mechanism underlies this process? It has been proposed that they are evolutionarily linked with each other and that recombination between distantly related elements leads to the emergence of novel mobile genetic elements[2]. Indeed, in prokaryotic mobile genetic elements, massive gene exchange between phages, transposons and plasmids occurred, making the borders of different classes of elements ambiguous[3]. The existence of network-like evolutionary relationships has also been proposed for eukaryotic mobile genetic elements[2, 4]. One good example is the transition from LTR retrotransposons to retroviruses and vice versa[5–7]. In addition, phylogenetic analyses imply that, for other elements, the transition events between transposon and virus occurred in the distant past[2, 8, 9]. However, due to the low sequence similarity between diverged mobile elements and the incongruence of phylogeny for each gene[10, 11], in most cases it remains ambiguous which element is actually a result of recombination events, leading to a transition in life cycle. Here, we report the analysis of a DNA transposon "Teratorn" in a small teleost fish medaka (Oryzias latipes), and provide direct evidence for genetic interaction between different classes of mobile elements, a fusion of a cut-and-paste DNA transposon and DNA virus.

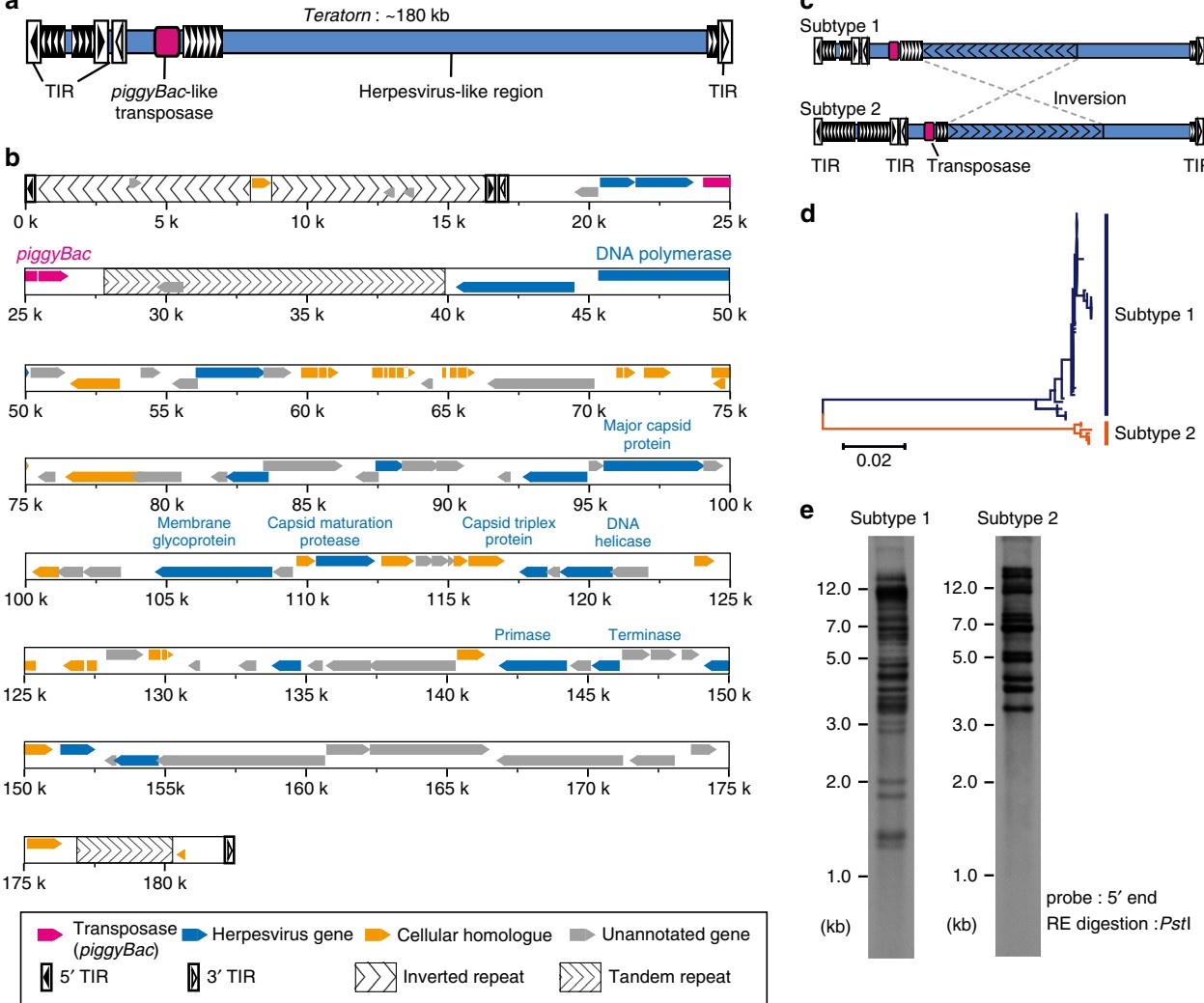

**Fig. 1** Sequence characteristics of Teratorn. **a** The overall structure of Teratorn. **b** Gene map of the subtype 1 Teratorn copy (73I9; named from the BAC clone ID). Predicted genes are classified into four categories depicted by colored arrowheads; magenta, piggyBac-like transposase gene; blue, herpesvirus-like genes; yellow, cellular homologues; gray, unannotated genes. Terminal inverted repeats (TIRs) of piggyBac-like transposon are depicted by boxed triangles. **c** Comparison of the whole structure of subtype 1 and subtype 2 Teratorn. **d** Neighbor-joining tree of all Teratorn transposase copies in the genome of medaka Hd-rR inbred strain. Note that they are separated into two clusters. The scale bar represents the number of substitutions per site. **e** Southern blot displaying individual Teratorn insertions in the genome of medaka Hd-rR inbred strain. The ~300-bp 5′ terminal region of Teratorn of each subtype was used as hybridization probes

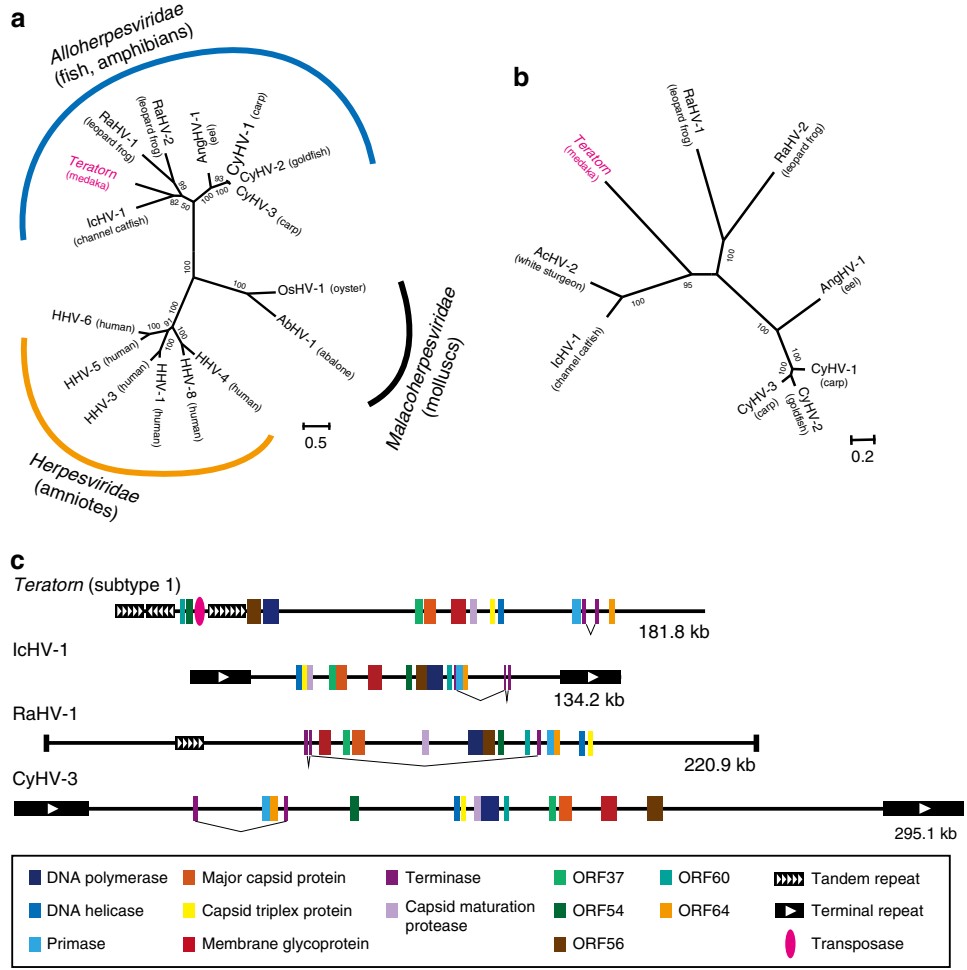

**Fig. 2** *Teratorn* contains a full herpesvirus genome. **a** Maximum-likelihood tree based on the amino-acid sequence of the DNA packaging terminase gene, the only gene confidently conserved among all herpesvirus species. Bootstrap values of branching are indicated at the nodes. Note that, among the three families (*Herpesviridae*, *Alloherpesviridae*, and *Malacoherpesviridae*) in the *Herpesvirales* order, *Teratorn* belongs to the family *Alloherpesviridae* (infecting fish and amphibians). **b** Maximum-likelihood tree based on the concatenated amino-acid sequences of major capsid protein, DNA helicase, DNA polymerase, and DNA packaging terminase from all alloherpesvirus species with sequenced genome. Within *Alloherpesviridae*, *Teratorn* is only distantly related to any other species. **c** Comparison of genomic structure of subtype 1 *Teratorn* and several alloherpesvirus species, as representatives of genera in the family *Alloherpesviridae*. The *colored squares* indicate each herpesvirus core gene, and *black boxes* depict repeat sequences. Note that *Teratorn* contains all 13 core genes conserved among all alloherpesvirus species, as well as long repeats. The *scale bars* in **b** and **c** represent the number of substitutions per site

This transposon was originally identified as an inserted DNA element in the medaka spontaneous mutant *Double anal fin* (*Da*)[12]. It was initially named *Albatross*, but recently renamed *Teratorn* (after the extinct group of very large birds of prey) to avoid confusion with a gene of the same name encoding a Fas-binding factor[12]. In our previous study, partial sequencing suggested that *Teratorn* is a DNA transposon that moves in a "cut-and-paste" manner, since it contains 18-bp terminal inverted repeats (TIR) at its termini and is flanked by target site duplications (TSDs). Interestingly, *Teratorn* was unusual in its relatively large size (at least 41-kb long[12]) for a transposon[13, 14]. We supposed that *Teratorn* is a giant DNA transposon with a unique evolutionary origin, life cycle and impact on the host genome. However, due to the presence of multiple copies as repetitive sequences, we failed to obtain the entire structure in the published draft genome constructed by shotgun sequencing[15].

In this study, we perform bacterial artificial chromosome (BAC)-based full-length sequencing of several *Teratorn* copies and find that *Teratorn* is 180-kb long, which is significantly larger than any other transposon reported so far. We identify a transposase gene homologous to *piggyBac* superfamily DNA

transposons, and detect its transposition activity both in vitro and in vivo. Surprisingly, *Teratorn* also contains the full genome of a herpesvirus, related to the family *Alloherpesviridae*, which infects fish and amphibians. In addition, genomic survey suggests that *piggyBac*-herpesvirus fusion also took place in yellow croaker (*Larimichthys crocea*) and nile tilapia (*Oreochromis niloticus*), implying the generality of transposon-herpesvirus fusion. We propose that *Teratorn* was created by a fusion of transposon and herpesvirus and as such, acquired a "bivalent" life form, behaving both as a transposon and a virus, assuring long-term survival in the host.

## Results

**_Teratorn_ is ~180-kb long**. We screened a BAC library of the Hd-rR strain to obtain clones containing the entire *Teratorn*. Using PacBio long-read sequencing we determined the full-length sequences of six *Teratorn* individual copies (Supplementary Fig. 1a) and found that in five clones, *Teratorn* was ~ 180-kb and in one clone it was ~155-kb (Fig. 1a, b, Supplementary Fig. 1b, c. Supplementary Data 1). The 155 kb clone is a shorter version of

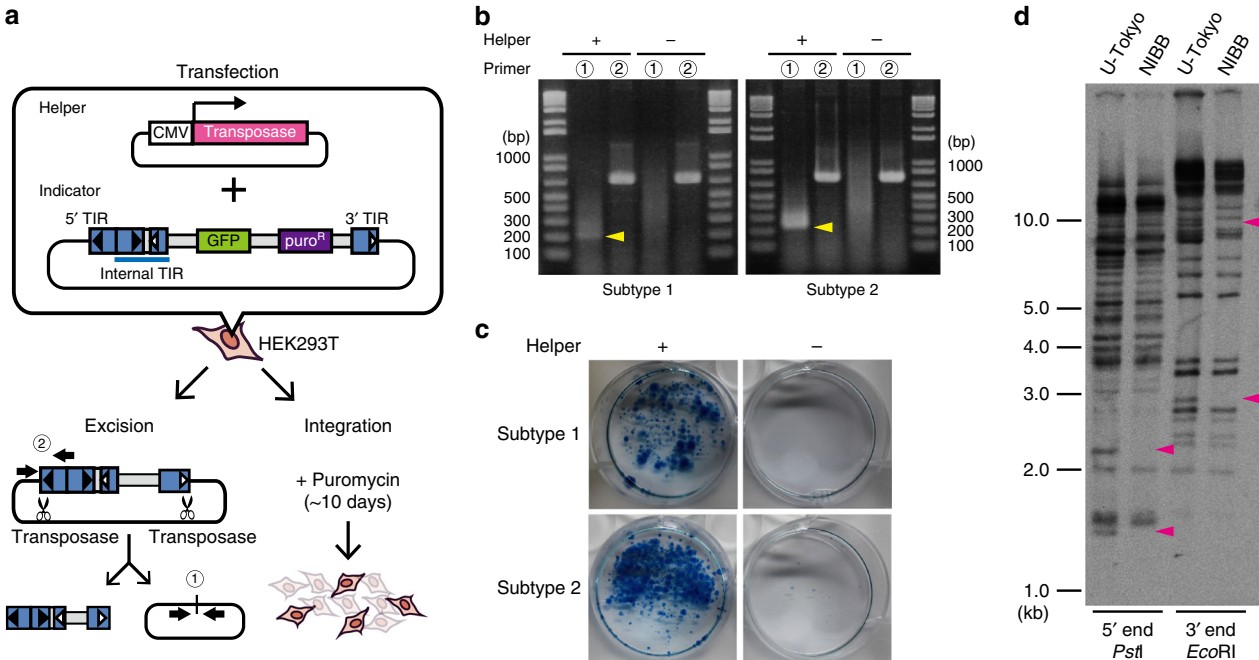

**Fig. 3** *Teratorn* retains transposition activity. **a** A schematic of the transposition assay. In the helper plasmid, *Teratorn* transposase gene was expressed under the CMV promoter. In the indicator plasmid, a GFP reporter and a puromycin-resistant gene were flanked by the 5′ and 3′ TIR. Note that additional TIR was inserted at the boundary of 5′ TIR and GFP cassette so as to mimic the endogenous *Teratorn* structure (internal TIRs). Transposition activity was examined by co-transfection of those two plasmids into HEK293T cells, followed by either PCR-based detection of transposon cassette excision from the indicator plasmid (excision assay, *left*) or chemical selection of transgenic cell lines (integration assay). *Thick arrows* indicate primer pairs used for the excision assay. **b** Excision assay. "+" and "−" indicate the presence and absence of helper plasmid, respectively. The number indicates the target region of PCR depicted in **a** (1, flanking the transposon cassette, amplifying only when excision reaction takes place; 2, targeting a terminus of transposon cassette, positive control). Note that PCR product flanking the transposon cassette was detected only when the helper plasmid was co-transfected (subtype 1, 202 bp; subtype 2, 250 bp; *arrowheads*). **c** Integration assay. Thirteen days after 7.5 μg/ml of puromycin selection, colonies were stained with methylene blue. Note that multiple colonies were observed only when the helper plasmid was co-transfected, indicating that the transposon cassette was integrated into chromosomes of HEK293T cells via transposition. **d** Southern blotting detecting individual *Teratorn* insertions in the Hd-rR individuals kept in the University of Tokyo (U-Tokyo) and at the National Institute for Basic Biology (NIBB), using sequences of *Teratorn* 5′ and 3′ ends as hybridization probes. Note that band patterns are different between the two individuals (*arrowheads*)

the other five with a partial deletion. In addition, we identified another subtype of *Teratorn*, which exhibits ~88% sequence identity but little similarity in the repetitive region including terminal region, by blast search of the full-length *Teratorn* sequence against a new, higher quality, version of the medaka genome (Fig. 1c, d, Supplementary Fig. 2) (http://utgenome.org/medaka_v2/#!Top.md). Structurally, the two subtypes are very similar to each other. Except for an 80-kb long inversion in the middle, the overall structure, such as the size, gene order (see below) and position of long repeats, is conserved (Fig. 1b, c, Supplementary Fig. 2). Thus, we concluded that they were derived from a common ancestor, and referred to the initially identified copies as *Teratorn* "subtype 1" and the second one as *Teratorn* "subtype 2". Blast search and southern blot analysis of their terminal sequences suggested that there exist 30–40 copies of subtype 1 and ~5 of subtype 2 per haploid genome of the medaka Hd-rR inbred strain (Fig. 1e).

***Teratorn* belongs to the *piggyBac* superfamily.** We first looked for a gene encoding transposase and found a gene homologous to *piggyBac* transposase inside *Teratorn* (magenta arrow in Fig. 1b for subtype 1, Supplementary Fig. 2a for subtype 2). *piggyBac* is one of the major "cut-and-paste" DNA transposon superfamilies and is widely distributed among eukaryotes, especially in metazoans[13]. The transposase gene of *Teratorn* exhibits high sequence similarity with other *piggyBac* superfamily transposase genes with essential amino-acid residues strictly conserved,

e.g., the catalytic domain (DDD motif), especially the four aspartic acid residues required for the transposition reaction (Supplementary Fig. 3a)[16]. Furthermore, the sequence composition of TIRs and TSDs at its termini follows the rule of the *piggyBac* superfamily; *piggyBac* transposons contain TIRs with 12–19 bp, beginning with a "CCYT" motif, and preferentially target TTAA (Supplementary Fig. 3b)[13, 14]. These characteristics indicate that *Teratorn* belongs to *piggyBac* superfamily. *Teratorn* is thus the biggest among known transposons, as the largest transposons reported so far were a *Polinton*-like DNA transposon (up to 25-kb long)[8, 17] and a *Gypsy*-like LTR retrotransposon (up to 25-kb long)[18].

***Teratorn* contains the genome of a herpesvirus.** In addition to the transposase gene, GENSCAN[19] and GeneMarkS[20] predicted ~90 putative genes, covering more than 60% of *Teratorn* (Fig. 1b, Supplementary Fig. 2a, Supplementary Tables 1 and 2, Supplementary Data 2). Remarkably, among the predicted genes, 17 genes show sequence similarity to those of herpesviruses (*E*-value < 0.01; *blue arrows* in Fig. 1b, Supplementary Fig. 2a, Supplementary Tables 1 and 2). Those include genes required for virus propagation, such as DNA replication (DNA polymerase, primase and UL21 homolog DNA helicase), virion maturation (capsid maturation protease), packaging of viral DNA (large subunit terminase) and virus structural proteins (major capsid protein, subunit 2 capsid triplex protein and envelope glycoprotein), and are known to be essential for the life cycle of herpesviruses.

Besides the essential genes, ~20 genes show no sequence similarity to those of other herpesviruses but are found in other organisms, which may have been secondarily obtained from infected host genomes or other sources such as bacteria and viruses (*yellow arrows* in Fig. 1b, Supplementary Fig. 2a,

Supplementary Tables 1 and 2). Intriguingly, most of them appear to function in regulation of immune response or cell proliferation in hosts. The former includes cell surface proteins (CD276 antigen-like, CXCR-like, TNFR-like) and immune signal transduction factors (ZFP36-like, TARBP-like), while the latter

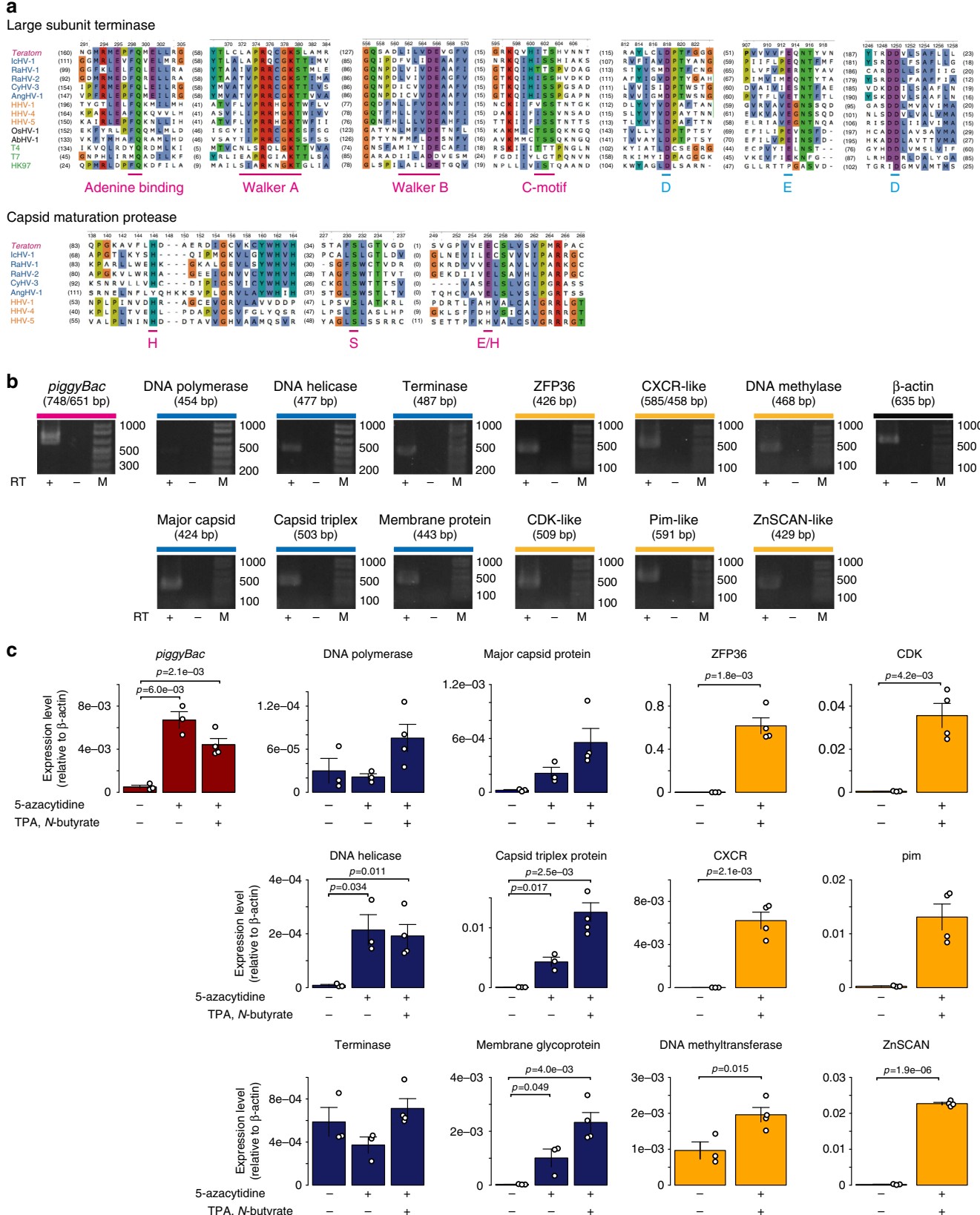

includes cell cycle regulator (CDK-like), apoptosis inhibitor (Mcl1-like), and oncogene (pim-like). The existence of these gene repertoires is a common feature of herpesvirus species that infect the hematopoietic cell lineages (like beta herpesviruses and gamma herpesviruses of amniotes)[21]. The other ~50 genes (*gray arrows* in Fig. 1b, Supplementary Fig. 2a) show no significant blast hit against the current non-redundant protein database. Given that large DNA viruses tend to carry many unknown genes in addition to essential ones[22, 23], it is likely that all predicted genes present in *Teratorn* (except for the tranposase gene) constitute a whole herpesvirus genome.

Herpesviruses are double-stranded DNA viruses that infect a wide variety of vertebrates and some invertebrates. Their genome is relatively large, ranging from 124-kb to 295-kb, and contains 70–200 genes[24, 25]. They share common characteristics, such as the way of DNA replication and packaging into the capsid, the three-dimensional structure of the capsid and the ability to establish a life-long persistent infection as a viral episome floating in the nucleus[21]. Herpesvirus species are classified into three families: *Herpesviridae* (amniotes), *Alloherpesviridae* (fish and amphibians), and *Malacoherpesviridae* (molluscs)[24]. Phylogenetic analysis based on amino-acid sequences of the terminase gene, the only gene confidently conserved among all herpesvirus species, suggests that *Teratorn* belongs to the family *Alloherpesviridae* (Fig. 2a). Within *Alloherpesviridae*, the same topology was obtained from the phylogenetic analysis using the concatenated amino-acid sequences of four genes (terminase, DNA polymerase, DNA helicase and major capsid protein, Fig. 2b). Importantly, *Teratorn* contains all 13 genes that are conserved among all alloherpesvirus species, and also shares other genomic features with alloherpesviruses such as the genome size, number of genes, GC content, and existence of long repeats (Fig. 2c, Supplementary Fig. 4)[24, 25]. All these data support the idea that *Teratorn* contains the complete genome of an alloherpesvirus.

**Teratorn exists as a fusion of *piggyBac* and herpesvirus.** The overall structure of the six *Teratorn* copies sequenced suggested that *Teratorn* is a fusion of the *piggyBac*-like DNA transposon and an alloherpesvirus. This raised the question of whether these two mobile genetic elements exist only in the fused form or also exist separately in the medaka genome. To address this question, we searched for genomic neighborhoods in all contigs from the Hd-rR genome assembly, which contain the transposase gene (Supplementary Fig. 5a). Among the 70 obtained contigs, all but one included the above herpesvirus genes at either 5′ or 3′ side to the transposase gene. For the remaining one contig, we were unable to examine the presence of viral genes because the assembly was interrupted by tandem repeats (Supplementary Fig. 5b). Thus, essentially no *Teratorn*

copy exists as a typical configuration of "cut-and-paste" DNA transposons (the transposase gene and TIRs). Furthermore, no herpesvirus-like sequences exists alone in the medaka genome. Taken together, we conclude that all *Teratorn* copies exist as a fusion of the *piggyBac*-like transposon and herpesvirus genome (Fig. 1a, b).

**Teratorn retains transposition activity.** The above structural characteristics suggest that *Teratorn* has at least the transposon-like life cycle. To test this idea, we first examined whether the encoded transposase is active utilizing an in vitro assay that directly detects transposition activity (Fig. 3a–c, Supplementary Figs. 6–8)[26–28]. In this assay, using HEK293T cells, we co-transfected helper plasmid that ubiquitously expresses transposase, and indicator plasmid that includes the puromycin-resistant gene flanked by the *Teratorn* terminal sequences, and tested excision and chromosomal integration of the transposon cassette (Supplementary Fig. 6a, c). Excision of the transposon cassette from the indicator plasmid was examined by PCR using primers that flank the transposon cassette. We found that a PCR band was detected only when the helper plasmid was co-transfected with the indicator plasmid (Supplementary Fig. 6a, b). We then tested the integration activity of *Teratorn* transposase. We cultured transfected cells in puromycin-containing medium for about 2 weeks, to screen for cells that became puromycin-resistant after chromosomal integration of the cassette. However, few colonies were detected after puromycin selection (Supplementary Fig. 6c, d).

Since *Teratorn* possesses the additional TIRs at the boundary of a pair of long inverted repeats and a unique region, i.e. 'internal TIRs' (Fig. 1a–c, Supplementary Fig. 2a), we hypothesized that internal TIRs are also required for the integration reaction. Thus, we inserted it into the indicator plasmid so as to mimic the endogenous structure of *Teratorn* and performed the same assays again (Fig. 3a). We found that both excision and integration reaction took place this time (Fig. 3a–c). For the excision assay, sequencing of PCR products revealed that the transposon cassette was precisely excised at the boundary, which is the characteristics of *piggyBac* superfamily DNA transposons[28] (Supplementary Fig. 7). For the integration assay, southern blot and sequencing analyses[29] of genomic DNA of surviving colonies confirmed the chromosomal integration of the plasmid sequence via transposition, although all of the copies sequenced so far were integrated via internal TIRs (Supplementary Fig. 8). Since DNA cleavage reaction could occur both at external TIRs and internal TIRs in the artificial circular form of the indicator plasmid (Supplementary Fig. 8b), this data implies that, for integration reaction, external TIRs are less frequently used than internal TIRs for unknown reasons. In any case, these results indicate that the

**Fig. 4** *Teratorn* encodes intact herpesvirus genes. **a** Multiple alignment of amino-acid sequences around catalytic centers of DNA packaging terminase and capsid maturation protease gene in *Teratorn* (*magenta*), herpesvirus species of *Alloherpesviridae* (*blue*), *Herpesviridae* (*orange*), *Malacoherpesviridae* (*black*) and bacteriophages (*green*). Note the conservation of catalytic residues of terminase (walker A, walker B, C-motif and adenine-binding motif in the ATPase domain (*magenta*), catalytic triads Asp-Glu-Asp in the nuclease domain (*blue*)), as well as the catalytic triad of protease (His-Ser-His/Glu; *magenta*)[32] in *Teratorn*. **b** RT-PCR of *Teratorn* genes in 5 dpf (days post fertilization) medaka embryos. "+" and "−" indicate that the reverse-transcription reaction was carried out or not, respectively. Cycle number of RT-PCR was 40. The *colors* indicate the categories of genes shown as in Fig. 1 (*red*, *piggyBac* transposase; *blue*, herpesvirus genes with known funtion; *yellow*, cellular homologues that seem to be involved in evasion of host immunity (ZFP36-like, CXCR-like and DNA methyltransferase-like) and cell proliferation (CDK-like, pim-like and ZnSCAN-like). **c** qPCR analysis of *Teratorn* genes in medaka fibroblast cells administered with or without 2 μM of 5-azacytidine, 3 mM of *N*-butyrate and 500 ng/ml of 12-*O*-Tetradecanoylphorbol 13-acetate (TPA). "+" and "−" indicate that each chemical was administrated or not. The value indicates the ratio of molar concentration relative to β-actin. Note that expression levels of most genes were moderately increased by chemical administration, although the expression level was still low. Statistical significance was tested by one-sided Welch Two Sample *t*-test. Each data point indicates the raw value of each experiment, and *bars* represent the mean ± SEM of replicates. Number of biological replicates are as follows; $n = 3$ for no chemical treatment, $n = 3$ for 5-azacytidine treatment, $n = 4$ for 5-azacytidine, TPA and *N*-butyrate treatment

**Table 1 Number of variants at the core herpesvirus genes among all subtype 1 *Teratorn* copies in the medaka genome**

| | Subtype 1 | | | | | |
|---|---|---|---|---|---|---|
| | Frameshift | Stop gained | Inframe deletion | Inframe insertion | Nonsynonymous | Synonymous |
| *piggyBac* | 0 | 1* | 0 | 1 | 89 | 54 |
| DNA polymerase | 0 | 1* | 0 | 0 | 72 | 96 |
| DNA helicase | 0 | 1* | 0 | 0 | 13 | 21 |
| Primase | 0 | 2* | 1 | 0 | 22 | 32 |
| Terminase1 | 0 | 0 | 0 | 0 | 15 | 13 |
| Terminase2 | 0 | 0 | 0 | 0 | 12 | 6 |
| Major capsid | 0 | 1* | 0 | 0 | 43 | 62 |
| Capsid triplex | 0 | 1* | 0 | 0 | 3 | 4 |
| Membrane protein | 0 | 1* | 0 | 0 | 53 | 69 |
| Protease | 0 | 0 | 0 | 0 | 33 | 30 |
| ORF37 | 0 | 1* | 0 | 0 | 32 | 33 |
| ORF54 | 0 | 4* | 0 | 1 | 78 | 46 |
| ORF56 | 0 | 1* | 0 | 0 | 119 | 126 |
| ORF60 | 1*** | 0 | 0 | 0 | 32 | 24 |
| ORF64 | 0 | 0 | 0 | 0 | 24 | 22 |

*Variant frequency ($p$) < 0.25; **0.25 < $p$ < 0.75; ***$p$ > 0.75

**Table 2 Number of variants at the core herpesvirus genes among all subtype 2 *Teratorn* copies in the medaka genome**

| | Subtype 2 | | | | | |
|---|---|---|---|---|---|---|
| | Frameshift | Stop gained | Inframe deletion | Inframe insertion | Nonsynonymous | Synonymous |
| *piggyBac* | 0 | 1** | 0 | 0 | 114 | 68 |
| DNA polymerase | 0 | 0 | 0 | 0 | 14 | 11 |
| DNA helicase | 0 | 1* | 0 | 0 | 3 | 7 |
| Primase | 0 | 0 | 0 | 0 | 11 | 9 |
| Terminase1 | 0 | 0 | 0 | 0 | 1 | 3 |
| Terminase2 | 0 | 0 | 0 | 0 | 3 | 0 |
| Major capsid | 0 | 0 | 0 | 0 | 0 | 1 |
| Capsid triplex | 0 | 0 | 0 | 0 | 3 | 6 |
| Membrane protein | 0 | 0 | 0 | 0 | 8 | 10 |
| Protease | 0 | 0 | 1 | 0 | 7 | 6 |
| ORF37 | 1*** | 0 | 0 | 0 | 4 | 10 |
| ORF54 | 0 | 0 | 0 | 0 | 15 | 7 |
| ORF56 | 1*** | 0 | 0 | 0 | 54 | 59 |
| ORF60 | 0 | 0 | 0 | 0 | 6 | 9 |
| ORF64 | 0 | 0 | 0 | 0 | 11 | 4 |

*Variant frequency ($p$) < 0.25; **0.25 < $p$ < 0.75; ***$p$ > 0.75

transposase of *Teratorn* is capable of mediating transposition in vitro.

To examine the transposition of *Teratorn* in vivo, we searched for integration site polymorphisms between the two groups of the Hd-rR medaka inbred strains, which had been kept separately for more than 20 years in our laboratory (University of Tokyo) and at the National Institute for Basic Biology. Southern blot analysis of *Teratorn* terminal sequences showed distinct band patterns between the two groups (Fig. 3d), suggesting that endogenous *Teratorn* transposition indeed occurred in vivo. Taken together, these results indicate that *Teratorn* still retains the activity of transposition in vivo and adopts the life cycle of the *piggyBac* family DNA transposon.

**Teratorn includes intact herpesvirus genes.** *Teratorn* is an active transposon, but could also retain some aspects of the herpesvirus. Indeed, *Teratorn* contains genes involved in virion morphogenesis (major capsid protein, capsid triplex protein, DNA packaging terminase, and capsid maturation protease, Figs. 1b and 2c, Supplementary Tables 1 and 2). Importantly,

sequence comparison revealed that catalytic sites in the ATPase and nuclease domain of terminase[30, 31], as well as the catalytic triad of capsid maturation protease[32], are conserved for *Teratorn* (Fig. 4a, Supplementary Fig. 9). For major capsid protein and capsid triplex protein, moderate sequence similarity to other alloherpesvirus species was detected (Supplementary Fig. 10, *blue*). Although there is little sequence similarity to *Herpesviridae* family, similar pattern of secondary structure (α-helix and β-sheet) was detected, suggesting the conserved three-dimensional structure of major capsid protein (Supplementary Fig. 10, *orange*)[33, 34]. These data suggest that *Teratorn* equips virion morphogenesis machinery.

To know whether *Teratorn* is still active as a virus, we examined the extent of open reading frame (ORF) degradation in *Teratorn* copies. Since Illumina whole-genome shotgun-sequencing data are publically available for the Hd-rR strain[35], which include sequences of all *Teratron* copies, we called SNPs and indels inside *Teratorn* by aligning *Teratorn*-derived reads to the *Teratorn* reference sequences. The reference sequences of subtype 1 and subtype 2 were constructed based on sequences from a single BAC clone (73I9) and on the consensus sequences deduced

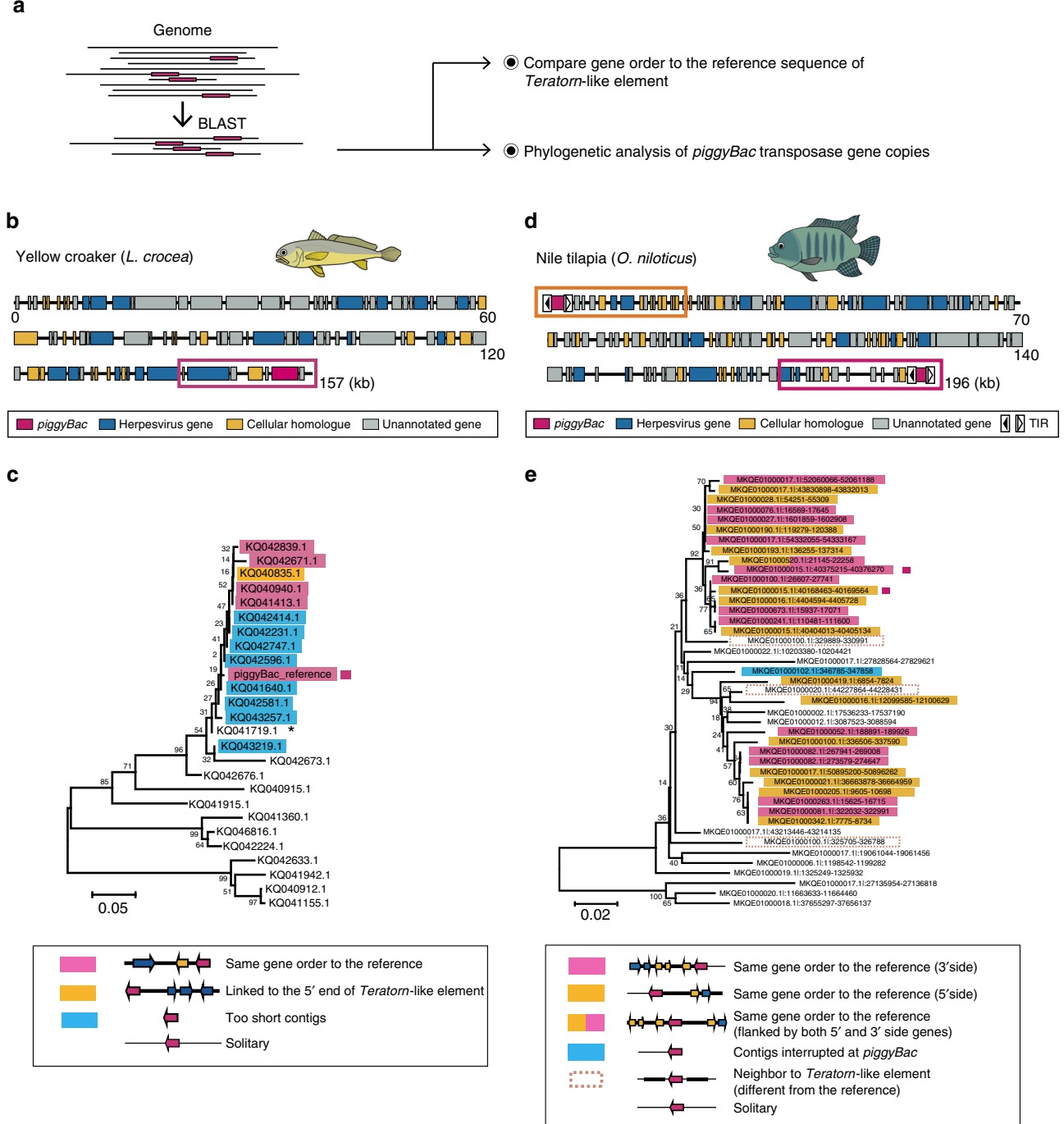

**Fig. 5** *piggyBac*-herpesvirus fusion in other species. **a** The procedure of screening all scaffolds that include the *piggyBac* transposase gene inside *Teratorn*-like element in yellow croaker (*Larimichthys crocea*) and nile tilapia (*Oreochromis niloticus*). First, contigs that contain the transposase gene were screened from the genome by blastn. For all contigs obtained, genomic neighborhoods around the transposase genes were tested by displaying alignment with the reference sequence of *Teratorn*-like element. In parallel, the phylogenetic relationship of the transposase copies was analyzed. **b** Reference sequence of *Teratorn*-like elements of yellow croaker. Predicted ORFs (exons) are depicted by *colored arrows* according to the categories; *magenta*, *piggyBac* transposase; *blue*, herpesvirus genes; *yellow*, cellular homologues; *gray*, unannotated genes. **c** Neighbor-joining tree based on the sequences of *piggyBac* transposase copies obtained by blast search is displayed. One region (1369–1598) of the reference transposase sequence was utilized for the phylogenetic tree construction. Each sequence is named by the contig name (e.g., KQ042839.1). *piggyBac* copies are categorized into four groups, according to the linkage to the herpesvirus genes. Note that *piggyBac* copies adjacent to the herpesvirus-like sequence (*magenta*) are clustered together, suggesting that a particluar type of *piggyBac* element is fused with the herpesvirus-like sequence. **d** Reference sequence of the *Teratorn*-like element of nile tilapia. **e** Neighbor-joining tree based on the sequences of *piggyBac* transposase copies obtained by blast search is displayed. *piggyBac* copies are categorized into six groups as described in the *inset*, according to the linkage to the herpesvirus-like sequence. Note that there are multiple *piggyBac* copies linked to the herpesvirus-like sequence in the same configuration as the reference sequence (either 5′ and 3′ end of *Teratorn* like element, *yellow* and *magenta*), although phylogenetically polyphyletic. The *scale bars* in **c** and **e** represent the number of substitutions per site

from all copies in the Hd-rR genome, respectively. We found that the number of nonsense mutations is much smaller than that of missense and synonymous mutations in coding regions (Tables 1 and 2, Supplementary Tables 3 and 4). Consistently, for the six *Teratorn* copies sequenced so far, all ORFs were found to be intact, suggesting that genes of *Teratorn* are currently functional.

Furthermore, we examined transcription of selected *Teratorn* genes including transposase and virus-related genes. Reverse transcription PCR (RT-PCR) analysis detected their ubiquitous and low levels of expression in nearly all tissues (e.g., brain, liver, muscles, gonads) and in developing embryos (5 dpf (days post fertilization)) (Fig. 4b, Supplementary Fig. 11). This pattern of expression could represent latency-associated conditions, and could change upon reactivation. To explore this possibility, we treated medaka embryonic-derived fibroblast cells with 5-azacytidine, *N*-butylate, and 12-*O*-Tetradecanoyl-phorbol 13-acetate (TPA), all of which are chemicals known to induce reactivation of latently infected herpesviruses in human cells[21, 36, 37]. We found that, although *N*-butyrate and TPA alone did not have an effect (Supplementary Fig. 12), some of the *Teratorn* genes were moderately derepressed after administration of 5-azacytidine; yet, the expression level of most genes was not sufficiently high ($10^{-5}$–$10^{-2}$ to $\beta$-actin, Fig. 4c), as compared with herpesvirus genes in activated mammalian cells[38].

Taken together, *Teratorn* encodes intact herpesvirus genes and thus likely utilizes the virus replication machinery for its propagation. However, conditions for full reactivation and virion formation remain elusive.

**Widespread colonization of *Teratorn* in the genus *Oryzias*.** To examine whether *Teratorn* is widely present in medaka genomes, we performed a genomic search against the genus *Oryzias*. There are more than 20 medaka-related species inhabiting Southeast Asia, from India to Japan. They are subdivided into three groups, *latipes* species group, *javanicus* species group, and *celebensis* species group (Supplementary Fig. 13a, b)[39]. PCR analysis of five herpesvirus core genes (DNA polymerase, DNA helicase, ATPase subunit of terminase, major capsid protein, and envelope glyco-protein) and transposase gene of *Teratorn* in 13 medaka-related species revealed that *Teratorn* exists in the *latipes* and *javanicus* species groups, but not in the *celebensis* species group (Supplementary Fig. 13b). Phylogenetic analysis suggests that the phylogeny of *Teratorn* and transposase genes is almost the same as that of the host species (Supplementary Fig. 13c). BAC screening and sequencing of the large part of *Teratorn* in the species of *javanicus* species group revealed that the configuration of *Teratorn*, particularly the location of *piggyBac* transposase, is conserved among *Oryzias* genus (Supplementary Fig. 13d, e, sequences and gene annotations are included in Supplementary Table 5 and Supplementary Data 2). These data demonstrate that *Teratorn* has widely colonized in the genomes of *Oryzias* genus, presumably using the *piggyBac* transposition machinery. Similar topology of phylogenetic trees between host genes and *Teratorn* genes implies that *Teratorn* was vertically descended from the common ancestor of *Oryzias* genus. However, considering the high sequence similarity among copies inside one species as well as the absence of degraded copies for all species except for *O. luzonensis*, we still cannot rule out the possibility that recent invasion of *Teratorn* into each species took place.

**Generality of *piggyBac*-herpesvirus fusion.** To further understand how general the *piggyBac*-herpesvirus fusion event might be, we performed comprehensive genomic survey of *Teratorn*-like sequences against all publically available vertebrate genome data. We found that *Teratorn*-like sequences were detected in several teleost fish species (19 of the 67 species, tblastn *E*-value < $10^{-3}$, more than 9 of the 13 alloherpesvirus core genes of *Teratorn*). Intriguingly, in several species (yellow croaker (*L. crocea*), nile tilapia (*O. niloticus*), ocean sunfish (*M. mola*) and turquoise killifish (*N. furzeri*)), we identified *piggyBac* transposase genes next to the *Teratorn*-like elements (Supplementary Fig. 14, sequences and gene annotations are included in Supplementary Tables 6 and 7 and Supplementary Data 2), suggesting that the fusion of the two mobile elements similarly to the one in medaka. To further explore this possibility, we focused on yellow croaker (accession: GCA_000972845.1)[40] and nile tilapia (accession: GCA_001858045.2) as these fish seem to have a high copy number of *Teratorn*-like elements, supported by multiple blast hits, compared with the other two fish (one remarkable blast hit for turquoise killifish and two hits for ocean sunfish, Supplementary Fig. 14). We searched for genomic neighborhoods in all scaffolds that harbor *piggyBac* transposase genes and compared to the reference sequence of the *Teratorn*-like element (Fig. 5a).

For yellow croaker, we obtained 40 scaffolds that include the *piggyBac*. Among the 17 obtained scaffolds that showed high sequence similarity with the reference *piggyBac* transposase (i.e., their *p* distance to the reference transposase sequence was < 0.041), five scaffolds contained the herpesvirus-like sequences next to the *piggyBac* transposase gene, in the same configuration as the reference shown in Fig. 5b. For 11 other scaffolds, we were unable to determine their neighborhoods because their length was too short. For one remaining scaffold, no herpesvirus-like sequence was found around the transposase gene, presumably caused by misassembly because the coverage of shotgun reads is peculiarly high at its flanking region (>100-fold to the transposase region). By contrast, for the remaining 23 contigs (their *p* distance was > 0.069), the transposase genes are not connected to any herpesvirus-like sequences. Indeed, phylogenetic analysis revealed that *piggyBac* copies adjacent to the herpesvirus-like sequence were clustered together (Fig. 5c). Thus, it is highly likely that the fusion of a particular *piggyBac* transposon and herpesvirus occurred in yellow croaker.

We next focused on nile tilapia (*O. niloticus*). Like in the analysis of yellow croaker, we searched for genomic neighborhoods for all copies of the same *piggyBac* and compared with the reference sequence of *Teratorn*-like element (MKQE0100015.1: 40163260–40398786), in which the herpesvirus-like sequence is flanked by the *piggyBac* sequences at both ends (Fig. 5d). This configuration was indeed observed in many of the identified elements (Fig. 5e). Importantly, in those elements, the regions flanked by the transposons are highly homologous, i.e., the region from the 5′ TIR of the 5′-side *piggyBac* to the 3′ TIR of the 3′-side *piggyBac*, while no sequence similarity was found outside (Supplementary Fig. 15), indicating the transposition of *Teratorn*-like element mediated by *piggyBac*. Therefore, we conclude that *Teratorn*-like element in nile tilapia is indeed the product of the fusion of *piggyBac* and herpesvirus, which allows for transposon-mediated propagation in the host genome. Taken together, these data suggest that the herpesvirus-*piggyBac* fusion event is not restricted to medaka, but frequently occurs in teleosts.

## Discussion

In the present study, we have characterized the mobile element *Teratorn*. Our study demonstrates that *Teratorn* has the gene encoding transposase of the *piggyBac* superfamily and retains transposition activity. One of the unique features of *Teratorn* is its size (~180-kb long), by far the biggest reported for a transposon. Thus, the transposition of *Teratorn* would be expected to have great impact on host genes and genomes. Indeed, among the previously published mutants, we found that at least four medaka

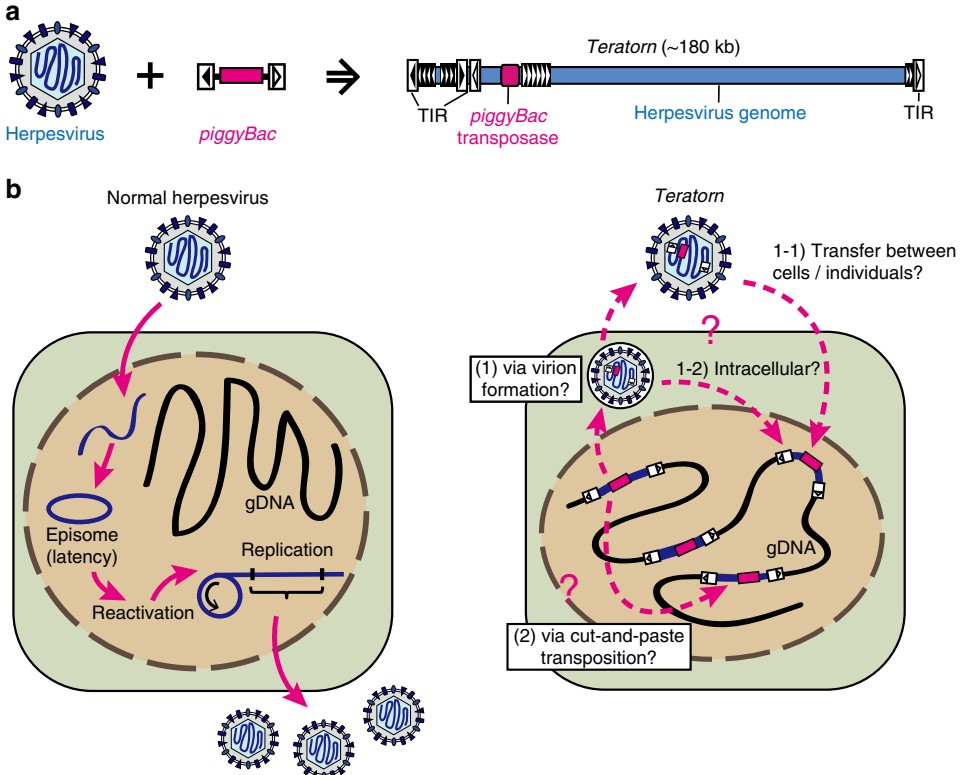

**Fig. 6** Model of *Teratorn* derivation from a herpesvirus that shifted to an intragenomic life cycle by gaining the *piggyBac* transposon system. **a** The entire structure of *Teratorn*. *Teratorn* is a fusion of a *piggyBac*-like transposon and a herpesvirus. **b** Comparison of life cycles of normal herpesviruses and *Teratorn*. Normal herpesviruses do not integrate their genome into chromosomes of host cells during their life cycles; instead, they form episomal DNA molecules and persist inside cells, accompanied by recursive reactivation (*left*). *Teratorn* might be derived from a herpesvirus that had shifted its life cycle to an intragenomic transposon-like parasite, by gaining the *piggyBac* transposon system. The transposition mechanism might be either via (1) DNA replication and virus particle formation or (2) conventional cut-and-paste transposition. Virus particles might be either (1–1) infectious and transmissible to other cells/individuals or (1–2) non-infectious and remain inside cells (*right*)

spontaneous mutants are caused by insertion of *Teratorn*; *rs-3* (*ectodysplasin-A receptor*)[41], *pc* (*glis3*)[42], *abc*[def] (*pkd1l1*)[43], and *Da* (*zic1/zic4*)[12] (Supplementary Fig. 16). Because of its serious impact, the transposition activity of *Teratorn* is likely suppressed in vivo, possibly through an epigenetic mechanism. Indeed, our recent study demonstrated that *Teratorn* inserted in the *zic1/zic4* locus is highly methylated[44]. Nonetheless, we detected insertion-site polymorphisms between the two long-separated groups of the medaka inbred line Hd-rR, suggesting that transposition of *Teratorn* occasionally occurs under natural conditions. Because of its size, *Teratorn* usually imposes deleterious effects on host gene function, but less frequently, can modulate the activity of long-range enhancers of a developmental gene, creating a novel trait in a host. One example is the medaka *Da* mutant (homozygous viable) in which *Teratorn* specifically disrupts the somite enhancers of the *zic1/zic4* locus without affecting the neural enhancer, leading to drastic changes in body shape and fins from larva to adult[12].

Surprisingly, *Teratorn* contains all essential genes shared by alloherpesvirus species with a similar genome configuration (Fig. 2c, Supplementary Fig. 4). We also found genes involved in the promotion of cell proliferation, inhibition of apoptosis, and immunomodulation, which is typical of lymphotropic herpesviruses. Importantly, in the majority of *Teratorn* copies, the above key genes are maintained intact and transcribed at low levels in medaka hosts. Based on these findings, we propose that *Teratorn* was descended from a herpesvirus that has propagated in the medaka genome by utilizing the *piggyBac* transposon system (Fig. 6).

Besides accidental chromosomal integration, there are only two endogenous herpesviruses, human herpesvirus 6 (HHV-6) and tarsier endogenous herpesvirus[45, 46], reported so far, out of > 100 herpesvirus species. They belong to the *Betaherpesviridae* and have an array of telomeric repeats (TMRs) at their termini, which are identical to the telomere sequences (TTAGGG)[45]. These herpesviruses are specifically integrated into the telomere region via homologous recombination through TMRs, presumably catalyzed by U94, an endonuclease-like gene homologous to parvovirus Rep[45, 47]. However, it remains unclear whether these endogenous viruses are genuine genomic parasites, because the tarsier endogenous herpesvirus has accumulated deleterious mutations in ORFs[46] and HHV-6 has not yet reached a fixation state among human populations (approximately 1% of the human population)[47]. In contrast, *Teratorn* has acquired the ability of transposition using its own transposase and colonized in the genomes of medaka species keeping the viral genes intact. Although the mechanism remains a mystery, one possible scenario is that an ancestral *Teratorn* was accidentally created by fusion of the two elements in other organisms, and the virus form of *Teratorn* then invaded into the medaka lineage. Since one of the hallmarks of herpesviruses is to establish a life-long persistent infection inside hosts as viral episomes without integration[21], we reason that chromosomal integration is an adaptive consequence to escape from host immunity and ensure a stable transmission of their progeny across host generations. The absence of genes involved in nucleotide metabolism in *Teratorn* (e.g., thymidine kinase (TK), deoxyuridine triphosphatase (dUTPase), and uracil DNA glycosylase), which are utilized for massive virus

propagation by modulating the nucleotide pool[21], is suggestive, because those genes might no longer be needed after endogenization. Taken together, *Teratorn* represents the first example of herpesvirus adaptation to intragenomic life cycle (Fig. 6).

The entire life cycle of *Teratorn* is still largely unknown, although it behaves at least as an active transposon in vivo. However, given that most herpesvirus-related genes also appear functional, those genes might be coordinately utilized for its propagation together with the transposase gene. Indeed, there is circumstantial evidence that chromosomally integrated HHV-6 (ciHHV-6) can be reactivated, e.g., in ciHHV-6-positive cultured cells[48], healthy ciHHV-6-positive individuals and X-linked severe combined immunodeficiency (X-SCID) patients[47]. Reactivation of these HHV-6 genes appeared to be accompanied by the formation of circular viral DNA molecules via excision of the telomeric t-loop[47, 49]. *Teratorn* could also undergo similar processes during reactivation; excision mediated by the *piggyBac*-like transposase followed by genome replication and virion formation (Fig. 6b). However, we have not succeeded in full activation of *Teratorn*. 5-azacytidine treatment of medaka fibroblasts only caused moderate reactivation of viral genes, and under this condition, we failed to detect virus proteins derived from *Teratorn* by western blot (major capsid protein, capsid triplex protein, and membrane glycoprotein, Supplementary Fig. 17). Given that all essential viral genes, especially genes involved in virion morphogenesis such as capsid proteins, DNA packaging terminase and capsid maturation protease, appear to be functional, it is still likely that, under as-yet-unknown conditions, *Teratorn* actually produces virus particles that infect new individuals. Similarly, virions have not yet been detected experimentally from *Polinton/Maverick* superfamily transposons, replicative large DNA transposons widespread among eukaryotic genomes[8, 17], although they contain a set of intact virus-like genes required for virion formation, including A32-like DNA packaging ATPase, Ulp1-like cysteine protease and two capsid proteins[9, 50].

Despite the potentially hazardous consequences of viral genes, there might be some benefits to the host having this large mobile element. One possible scenario is that *Teratorn* serves as a shield against an otherwise lethal virus, by inhibition of virus entry by receptor block, inhibition of functional virion assembly or establishment of immunotolerance. On the other hand, *Teratorn* shows characteristics of selfish genetic elements, since all herpesvirus essential genes and the transposase gene remain intact. Thus, we propose that *Teratorn* might serve two purposes: one is as a guardian against exogenous virus infection, and the other is as a selfish intragenomic parasite. Recently, it was shown that endogenous virophages in some protozoan genomes were reactivated by superinfection of giant viruses (preys of the virophages), which facilitates host survival and their own propagation[51, 52]. *Teratorn* could undergo a similar response during infection with exogenous viruses. In any case, further experimental efforts to detect virions will be necessary to understand the life cycle of *Teratorn* and the biological significance of the existence of *Teratorn* in the medaka genome.

As mentioned above, *Teratorn* is a mobile element that experienced the fusion between virus and cut-and-paste DNA transposon in eukaryotes. Importantly, our genomic survey implied that the herpesvirus-transposon fusion event is not restricted to medaka but is rather frequent among teleost lineages (Fig. 5, Supplementary Fig. 14). *Teratorn*, however, cannot be the only example in the network of eukaryotic mobile elements. Indeed, all viruses have the potential to shift into the intragenomic life cycle if they acquire an integration system from other sources. In this context, *Polintons* are of particular interest, since recent phylogenetic studies suggested that *Polintons* have evolutionary links with other mobile genetic elements

(adenovirus, virophages, linear plasmids, and bacteriophages) and serve as a hotbed for recombination leading to a change in their life cycle[8, 9, 11]. Furthermore, recombination events between distantly related viruses have been reported, such as chimeric viruses (chimera of single-stranded DNA (ssDNA) and single-stranded RNA (ssRNA(+)) virus) and bidnavirus (chimera of parvovirus, *Polinton*, reovirus, and baculovirus)[2, 9, 53, 54]. Thus, recombination and fusion between mobile elements of distinct classes occur more frequently than previously thought, leading to the diversification of mobile genetic elements. Further identification of novel and peculiar mobile genetic elements will provide new insights into mechanisms underlying their diversification and evolution.

## Methods

**Fish strains**. Hd-rR, d-rR, and HNI inbred strain of *Oryzias latipes* were maintained in our laboratory. Fish strains of medaka-related species were obtained from the laboratory stocks maintained at Niigata University and National Institute for Basic Biology. Lists of each species and their original collection sites are described in Supplementary Table 8. All experimental procedures and animal care were performed according to the animal ethics committee of the University of Tokyo.

**Screening and sequencing of BAC clones**. *Teratorn* insertion sites were identified by screening of the 5′ and 3′ flanking regions by blastn of *Teratorn* terminal sequences against the public genome assembly of medaka Hd-rR inbred strain[15], followed by mapping of those obtained sequences to the genome of another medaka inbred strain, HNI, to ascertain which of the 5′ and 3′ flanking sequences are derived from the same loci. For the 12 *Teratorn* copies with identified integration sites, BAC clones that were PCR-positive for both *Teratorn* ends were screened from the medaka Hd-rR BAC library[55]. Sequences of the primers for screening are in Supplementary Table 9. For the screened six individual *Teratorn* copies, BAC DNA was purified by QIAGEN Large-Construct Kit (QIAGEN). Sequencing of BAC clones was implemented using PacBio RS-II. Assembly was carried out using HGAP pipeline.

For medaka related species (library name: IMBX for *O. dancena*; OHB1 for *O. hubbsi*; OJV1 for *O. javanicus*; LMB1 for *O. luzonensis*), BAC clones were screened by PCR of the four *Teratorn* genes (*piggyBac*-like transposase, membrane glycoprotein, DNA polymerase and DNA packaging terminase). Sequences of the primers for screening are in Supplementary Table 9. BAC DNA was then purified using NucleoBond® Xtra BAC (MACHEREY-NAGEL). The sequencing library was prepared by sonicating to a size ranging from 800 to 1200 bp, followed by adaptor ligation and amplification using KAPA Hyper Prep Kit (KAPA BIOSYSTEMS). Paired-end sequencing of the prepared libraries was executed on the Illumina Miseq platform. After filtering out low-quality reads by trimmomatic v0.33[56], de novo assembly was carried out by CLC Genomic Workbench (https://www.qiagenbioinformatics.com/).

**Gene annotation**. Gene annotation was initially carried out by the GeneMarkS web server[20]. For genes that start with codons other than "ATG", prediction by ORF finder (http://www.ncbi.nlm.nih.gov/gorf/gorf.html) was used instead. If adjacent multiple ORFs seemed to be derived from a single gene (i.e., different portions of the same gene were obtained as blastp output), gene annotation by GENSCAN web server[19] was used to generate a more plausible gene model including introns.

**Identification of subtype 2 *Teratorn***. Subtype 2 *Teratorn* was identified by blastn search of subtype 1 *Teratorn* sequence against the updated version of Hd-rR genome assembly (http://utgenome.org/medaka_v2/#!Top.md). The putative full-length subtype 2 *Teratorn* sequence was constructed by conjugating partial sequences of two contigs (ctg7180000008209: 1–30344, ctg7180000008207: 2815545–2982619), followed by calling consensus sequence by Bcftools[57], based on the Illumina whole-genome shotgun short reads (see below).

**Multiple alignment of active *piggyBac* transposase genes**. Amino acid sequences of *piggyBac* superfamily transposase genes (*piggyBac* in *T. ni*, *Uribo2* in *X. tropicalis*, *yabusame-W* in *B. mori*, *piggyBat* in *M. lucifugus*, *AgoPLE* in *A. gossypii*, and *PLE-wu* in *S. frugiperda*) were downloaded from GenBank. Those transposases were aligned using Clustal W with default parameters[58]. The multiple alignment was visualized using TrimAl[59].

**Phylogenetic analysis**. For the phylogenetic analysis of *Teratorn* among *Herpesvirales*, amino-acid sequences of DNA polymerase, DNA helicase and major capsid protein of *Alloherpesviridae* species, as well as DNA packaging terminase of *Herpesvirales* species and T4 phage, downloaded from GenBank, were used.

Multiple sequence alignment was built up using MUSCLE in MEGA6 package[60]. Poorly aligned regions were removed using trimAl with -strictplus option[59]. Maximum-likelihood analysis was carried out using MEGA6. Le_Gascuel_2008 model, considering evolutionary rate difference among sites by categorizing into five values to fit Gamma distribution, with a proportion of sites being invariable, was used as substitution model with 1000 bootstrap replicates.

For the analysis of Teratorn transposase copies in the Hd-rR genome, nucleotide sequences of all Teratorn transposase copies, screened from the genome assembly data by blastn, were used. Multiple sequence alignment was built up using MUSCLE. Neighbor-joining analysis was carried out using MEGA6. Jukes-Cantor model, considering uniform evolutionary rate among sites, was used as substitution model with 1000 bootstrap replicates.

For the analysis in the genus Oryzias, partial sequences of several Teratorn genes were screened from the genome of each species as described below. Multiple alignment of Teratorn genes and concatenated sequences of host genes were constructed by MUSCLE. Maximum-likelihood analysis was carried out using Tamura-Nei method as substitution model with 1000 bootstrap replicates, without considering evolutionary rate difference among sites.

**Search of neighboring region to the piggyBac transposase.** Contigs or scaffolds including the transposase gene were screened by blastn search of the transposase gene, followed by extraction of flanking region by BEDtools (upstream 25 kb and downstream 25 kb region for medaka, 60 kb flanking region for yellow croaker, and 20 kb flanking region for nile tilapia). Gene order around the transposase genes were then assessed by displaying dot plot matrix between the reference Teratorn-like element and the obtained contigs, using mafft on line server[61]. Neighbor-joining analysis were carried out based on the partial region of the transposase gene using Tamura-Nei method as substitution model with 1000 bootstrap replicates, without considering evolutionary rate difference among sites.

**Plasmid construction for the transposition assay.** Indicator plasmid was constructed as follows. First, the puromycin-resistant gene cassette was extracted from the pMXs-puro retroviral vector by digesting with BamHI and XhoI. In parallel, the AcGFP expression cassette was amplified by PCR. These two DNA fragments were then conjugated using in-fusion HD cloning kit (TaKaRa). This puro^R-AcGFP fragment was further conjugated with Teratorn TIRs at both sides by joint-PCR, and was subcloned into pCR-BluntII-TOPO (Invitrogen). Finally, Teratorn internal TIR was inserted into the boundary of the 5′ TIR and the puro^R-AcGFP cassette so as to mimic the endogenous Teratorn structure. Helper plasmid was constructed by RT-PCR of the Teratorn transposase followed by subcloning into pCSf107mT vector[62]. Sequences of the primers for the construction of the plasmids are in Supplementary Table 9.

**Transfection and excision and integration assay.** HEK293T cell line was a kind gift from Prof. Yoshinori Watanabe (University of Tokyo). Cells were maintained in Dulbecco's Modified Eagle's Medium (DMEM) with 10% FBS and 100U/ml penicillin and 100 μg/ml streptomycin (Gibco) at 37 °C with 5% $CO_2$. The day before transfection, cells were seeded in 6-well plates to achieve 70–90% confluent on the next day. Lipofectamine 3000 reagent (Invitrogen) was used to co-transfect cells with 1500 ng of helper plasmid and 2500 ng of indicator plasmid per well. For the excision assay, 40% of cells were passaged to a new dish, cultured for about 2 days to become fully confluent and the plasmid was extracted by the Hirt method[63]. PCR was carried out using primers that flank the transposon cassette of the indicator plasmid. Sequences of the primers for the excision assay are in Supplementary Table 9. In the integration assay, 40% of transfected cells were passaged to a new dish and cultured in DMEM with 5.0, 7.5, or 10.0 μg/ml of puromycin for about 2 weeks. Medium was changed every 2 or 3 days. Surviving colonies were isolated by peeling off and sucking up with a pipette and transferred to a new dish. When cell clones grew in sufficient number, genomic DNA was isolated using Wizard Genomic DNA extraction kit (Promega) for Southern blotting and PCR-based integration site determination.

**Southern blotting.** In all, 5–10 μg of genomic DNA extracted from medaka adult whole body or HEK293T cells were digested with appropriate restriction enzymes for at least one hour and precipitated with ethanol. Digested DNA was resolved by gel electrophoresis with 25–35 V overnight, and transferred to a Hybond-N+ nylon membrane (GE Healthcare). AlkPhos Direct Labeling and Detection System with CDP-Star (GE Healthcare) was used for hybridization and signal detection. Purified PCR products of Teratorn terminal sequences were used as hybridization probes to visualize individual Teratorn insertions. Sequences of the primers for the preparation of template DNA for probes are in Supplementary Table 9.

**Extension primer tag selection linker-mediated PCR.** Extension primer tag selection linker-mediated PCR ((EPTS)LM-PCR) was carried out as previously described[29]. First, 1.5 μg of genomic DNA from HEK293T cells was digested with HaeIII or ApoI overnight. After precipitation with ethanol, a biotinylated primer specific to Teratorn terminal sequence was used for tagging the transposon insertion sites. In this step, Phusion High-Fidelity DNA Polymerase (Thermo) was used for DNA extension (98 °C for 3 min, 68 °C for 30 min, and 72 °C for 30 min).

After purifying the PCR products using the Wizard® SV Gel and PCR Clean-Up System (Promega), biotin-tagged DNA was isolated by incubating with Dynabeads® M280 Streptavidin (Thermo) for one hour, followed by twice washing out of non-target DNA. Finally, double-stranded oligonucleotide phosphorylated linkers were ligated at 16 °C overnight. PCR with primer pairs specific to Teratorn terminal sequence and linker oligonucleotide were carried out to amplify Teratorn insertion sites. Sequences of the primers for this analysis are in Supplementary Table 9.

**Multiple alignment of herpesvirus genes.** Multiple alignment of amino acid sequences of DNA packaging terminase, capsid maturation protease, major capsid protein and subunit 2 capsid triplex protein were constructed by PROMALS3D[64]. Sequences of each species were downloaded from Genbank. Catalytic centers of terminase and protease gene was characterized based on Rao V.B. and Feiss M.[30], Selvarajan Sigamani S. et al.[31], and Cheng H. et al.[32]. Domains of major capsid protein was characterized based on Huet A. et al.[33].

**Variant calling and annotation.** Reference medaka genome data was reconstructed as follows, in an attempt to map all Teratorn-derived reads to the reference Teratorn sequences. Teratorn was masked from the public medaka draft genome[15] by blastn of both Teratorn subtypes, followed by maskFastaFromBed command of BEDtools. Then, the genome was conjugated with the sequences of Teratorn. Illumina whole-genome shotgun read data was downloaded from DDBJ Sequence Read Archives (accession: DRR002213). After filtering out low-quality bases and adapter sequences from short reads by trimmomatic v0.33[56], reads were aligned to the reconstructed reference genome using BWA (Burrows-Wheeler Aligner)-MEM[65], with the default parameter settings. Removal of PCR duplicates was carried out by Picard (http://picard.sourceforge.net). Local realignment was executed using RealignTargetCreator and IndelRealigner tools in GATK, with the default parameters[66]. Variant calling was performed by UnifiedGenotyper tool in GATK, with the following parameters (ploidy, 30 for Teratorn subtype 1 and 5 for subtype 2; stand-call-conf, 30; stand-emit-conf, 20; glm, BOTH). Variant annotation was implemented using SnpEff, with the default parameters[67].

**Expression analysis of Teratorn genes in medaka.** Total RNA was isolated from 5 dpf (days post fertilization) embryos and adult fish tissues (brain, liver, fin, muscle, gut, ovary, testis) of medaka Hd-rR strain, using ISOGEN (Nippon Gene), according to the manufacturer's protocol. After removal of genomic DNA by DNaseI digestion (Invitrogen), Complementary DNA (cDNA) was synthesized using SuperScript III (Invitrogen). RT-PCR was carried out using Phusion DNA Polymerase (Thermo Fisher) for 40 cycles. Sequence of the primers for RT-PCR are in Supplementary Table 9.

**Test of reactivation of Teratorn genes by chemicals.** Medaka embryonic fibroblast cell line was previously established in our laboratory. Cells were maintained in L-15 medium (Gibco), supplemented with 15% FBS (biosera), 100 U/ml penicillin and 100 μg/ml streptomycin (Gibco) at 30 °C. For reactivation, cells were administered with 2 μM of 5-azacytidine (Sigma) for 5 days. For a subset of samples, 500 ng/ml of 12-O-Tetradecanoylphorbol 13-acetate (Sigma) and 3 mM of sodium butyrate (Sigma) were also administered for the last 2 days. This experiment was repeated for three (no chemical administration, 5-azacytidine only) or four (TPA + N-butyrate, TPA + N-butyrate + 5-azacytidine) times. Total RNA extraction and cDNA synthesis were carried out as described above. Quantitative reverse transcription PCR (qRT-PCR) was carried out with the Stratagene mx3000p system (Agilent), using the THUNDERBIRD SYBR qPCR Mix (ToYoBo). β-actin was used as an internal control. Sequence primers for qRT-PCR are in Supplementary Table 9. Ratio of molar concentration of Teratorn genes relative to β-actin was quantified based on standard curves generated from purified PCR products, using Qubit® 2.0 Fluorometer (Thermo Fisher) to measure mass concentration.

**Statistics.** In the qRT-PCR experiment described above, no statistical method was used to predetermine sample size. Sample sizes were based on previously reported experiments that are similar to the present study. No data were excluded from the analysis. For genes that met the assumption of normal distribution (all genes except for major capsid protein, pim, and terminase, Shapiro-Wilk test, $p < 0.05$), statistical significance was tested by one-sided Welch Two Sample t-test. All statistical analyses were performed using R software (version 3.2.4).

**Antibodies.** Partial sequences of major capsid protein (amino acids 934–1074), capsid triplex protein (amino acids 51–170) and membrane glycoprotein (amino acids 711–830) were subcloned into pGEX4T-1 vector (GE Healthcare). After culture of transformed bacteria, GST-tagged protein fragment was purified using Glutathione Sepharose 4B (GE Healthcare). Polyclonal antibodies were raised by immunization of mouse with the GST-tagged truncated proteins (100 μg) for about 2 months, followed by extraction of serum. Sequences of the primers for antigen production are in Supplementary Table 9.

**Western blotting**. For reactivation of *Teratorn* genes, medaka fibroblast cells were administered with 5-azacytidine (0.0 µM, 1.0 µM, 1.5 µM, 2.0 µM) for 5 days. For a subset of samples, 500 ng/ml of 12-*O*-Tetradecanoylphorbol 13-acetate (Sigma) and 2 mM of sodium butyrate (Sigma) were also administered for the last 2 days. For positive control, HEK293T cells transfected with the plasmids that express the GFP-tagged antigen protein fragment were used. Cells were lysed with 1× SDS buffer, separated by polyacrylamide gels (5.5% for major capsid proteins and membrane glycoprotein, and 10.0% for capsid triplex protein, respectively) and transferred onto polyvinylidene difluoride membranes (Millipore). After blocking with 5% skim milk for 1 h at room temperature, membranes were incubated with primary antibodies (anti-major capsid protein, anti-capsid triplex protein, and anti-membrane glycoprotein, 1:2000 dilution) over night at 4 °C. After several washes, membranes were incubated with secondary antibody (Anti-Mouse IgG HRP-Linked Whole Ab Sheep (GE Healthcare, NA931V), 1:2500 dilution) for 1 h at room temperature. Protein bands were visualized using the ECL Select Western Blotting Detection Reagent (GE Healthcare) and detected by ImageQuant (GE Healthcare).

**Screening of *Teratorn* genes in the genus *Oryzias***. Partial sequences of five herpesvirus genes (DNA polymerase, DNA helicase, major capsid protein, membrane glycoprotein, and ATPase subunit of terminase) were screened from genomic DNA of each medaka related species by PCR, using degenerate primers constructed from the multiple alignment of *Teratorn*-like sequences in several teleost fish (*S. salar*, *O. mykiss*, *E. lucius*, *O. niloticus*, *P. nyererei*, *C. semilaevis*). For genes in which degenerate primers didnot work, primers designed from sequences of subtype 1 and subtype 2 *Teratorn* of *O. latipes* were used instead. *piggyBac*-like transposase gene was screened using primers that were designed from *O. latipes* genome. Sequences of the primers for the screening are in Supplementary Table 9.

**Search of *Teratorn*-like elements in other species**. Tblastn search of 13 herpesvirus core genes of *Teratorn* was carried out against all available teleost genomes with default parameters. In addition, tblastn of the four genes (DNA polymerase, DNA helicase, DNA packaging terminase, and major capsid protein) was performed against amphibians, chondrichthyes or sarcopterygi in the web browser. Contigs or scaffolds that include a series of herpesvirus-like sequences were screened as follows. First, location of the 13 herpesvirus core genes was identified by tblastn. After merging the genomic loci which are within 40 kb of one another, sequences of the defined region and the flanking 40 kb region were extracted from the draft genome by BEDtools.

**Consensus sequence calling of *Teratorn*-like elements**. In yellow croaker, preliminary consensus sequence was constructed using Unipro UGENE[68], based on multiple alignment of seven contigs that include the relatively long range region of *Teratorn*-like sequence (> 40 kb). Then the consensus sequence was created by mapping of the Illumina whole-genome shotgun reads (Accession: SRP041375), followed by consensus sequence calling by vcftools[57]. In nile tilapia, all gained *Teratorn*-like sequences were aligned to the longest one (MKQE01000015: 40163260-40398786) using BWA (Burrows-Wheeler Aligner)-MEM[65], followed by extraction of consensus sequence using Unipro UGENE.

**Data availability**. Nucleotide sequence data and gene annotation of *Teratorn* and *Teratorn*-like elements are provided and described within the Supplementary Data of this manuscript. Sequence of subtype 1 *Teratorn* (73I9) has been submitted to the DDBJ/EMBL/GenBank databases (Accession No: LC199500). All other data supporting the findings of this study are available from the corresponding author upon request.

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

## Acknowledgements

We thank Dr W. Qu for the help of identification of *Teratorn* integration sites, Drs. T. Kimura and Y. Takehana for the help of BAC screening, Drs. A. Kato and J. Arii for valuable discussion and experiments of reactivation of *Teratorn*, Prof. Deshou Wang and Mr. Xianbo Zhang for the help of fosmid screening to determine the sequence of *Teratorn*-like elements in nile tilapia, Dr A. Terashima for critical reading of the manuscript, and Mrs. I. Fukuda for fish care. The five *Oryzias* species (*O. sarasinorum*, *O. nigrimas*, *O. marmoratus*, *O. matanensis*, *O. profundicola*) used in the present study were provided by Niigata University through the National Bio-Resource Project of the Ministry of Education, Culture, Sports, Science, and Technology (MEXT), Japan. This work was supported by CREST, JST (Grant No: JPMJCR13W3).

## Author contributions

H.T. planned and supervised this research. Y.I., T.S., and K.N. screened BAC clones. S.M. identified *Teratorn* integration sites in the genome of Hd-rR strain and conducted BAC sequencing. T.S and M.K. performed sequencing of BAC clones of medaka related species. Y.I., A.S., and A.K. designed the experiments and performed transposition assay. Y.I and Y.K. designed the experiments and examined reactivation of *Teratorn*. Y.I., T.S., T.A., and M.K. performed phylogenetic analysis in *Oryzias* genus. Y.I., T.A., and M.K. performed data analysis. Y.I. and H.T. wrote the paper.

## Additional information

**Competing interests:** The authors declare no competing financial interests.

