## [Peer Review File · Nature Communications]

REVIEWERS' COMMENTS:

Reviewer #1 (Remarks to the Author):

Mobile genetic elements (MGE) are major players in genome evolution. Here, the authors report a novel MGE called Teratorn in the genome of the medaka. Teratorn is the product of a fusion between a herpesvirus and a piggybac transposase. The authors characterize two subtypes of Teratorn in the genome of the Hd-Rr medaka strain, demonstrate that the element is capable of both excision and integration in HEK293 cells and that it is present in all four local populations of medaka in varying copy numbers.

It was a pleasure to read this paper, the discovery of this new element is definitely exciting and intriguing. Overall, I found that the analyses performed by the authors are robust and that the results support most of the conclusions. Below is a list of comments that should however be addressed:

1-1) - based on the presence of Teratorn in all four populations of medaka, the authors conclude that the element was present in the ancestor of these populations and that it has been transmitted vertically over 25 million years. In my opinion, this conclusion is weak because it is only based on indirect evidence. The congruence between the phylogeny of Teratorn and that of the medaka gene could be due to the fact that Teratorn is transmitted horizontally (in addition to vertically) on a regular basis, and that such transfers tend to occur between medaka individuals that are closely related both phylogenetically and geographically. Such horizontal transmission is plausible given that genes of viral origin that are involved in viral particle formation and virus replication are intact (devoid of non-sense mutations). Teratorn may well be inherited both vertically and horizontally and its presence in the medaka species may be much more recent than 25 myrs.

A direct evidence demonstrating the proposed antiquity of Teratorn would be to identify orthologous Teratorn copies between the various medaka strains, i.e., copies that are present at the same locus in two, three or even four strains. More specifically, a definitive conclusion on the presence of Teratorn in the medaka genome for 25 myrs requires the characterization of at least one copy shared at an orthologous locus between the HSQK or Nilan populations and

at least one of the three other populations. Such a search for orthologous insertions requires a better characterization and comparison of the regions flanking each Teratorn copy in the four strains.

1-2) - Based on the presence of a Teratorn element showing the same gene order in both the javanicus and latipes species group of the genus *Oryzias*, the authors conclude that this element originated in the common ancestor of these species and that it was inherited vertically during 60 Myrs. It has been shown that once integrated in a host genome, copies of mobile elements transmitted vertically in host populations typically evolve neutrally (e.g. Lampe et al. 2003 MBE). If Teratorn is present in *Oryzias* spp since more than 60 Myrs, many old and highly degraded copies should be found in the genome of these species and the average pairwise distance between copies within a given genome should be rather high, contrary to what is observed (Page 9, Line 3). Furthermore, if Teratorn is old, as proposed by the authors, some copies should be shared at orthologous locus between species from the latipes and javanicus groups. I thus reiterate my request of performing a search for such orthologous copies. Without such direct evidence for vertical inheritance, the authors cannot conclude without doubt that the element was present in the ancestor of the *Oryzias* genus 60 Myrs ago. Much like in the first paper, this point should be further explored. How many stop codons are expected during 60 Myrs under neutral evolution in teleosts? Are copies of Teratorn showing this expected number of stop codons observed? Would an alternative hypothesis that would imply a more recent age for Teratorn and involve a mix of vertical and horizontal transfer not be more plausible?

2 - Teratorn, like other known genetic elements such as retroviruses and polintons, is at the boundary between viruses and transposable elements. Its transposase and terminal inverted repeats allow it to integrate and multiple itself into host genomes and its genes of viral origin may allow him to form infectious viral particles. In the discussion, the authors first conclude that Teratorn is a DNA transposon (p19, line 5) and then propose that it is a novel herpesvirus endogenized in the medaka genome (p21, line 7). This is confusing and may really impede the understanding of readers who are not familiar with the complexity of mobile genetic elements. The potentially dual nature of Teratorn should be more explicitly described and the authors should not attempt to provide a definitive conclusion on whether it is a virus or a transposon. They should also clearly mention that to be confirmed, the putative viral fonctions of Teratorn

await observation of viral particles.

3 - The capacity of Teratorn to reactivate and form viral particles is tested by the authors through treatment of medaka embryonic cells with chemicals that are known to reactivate latent human herpesviruses. No significant reactivation is observed. In a recent study, Fischer & Hackl (2016) show that the endogenous virophage Mavirus is reactivated by the presence of the Cafeteria roenbergensis virus, with which it shares promoter motifs. I was wondering whether the authors could have access to a strain of alloherpesvirus that they could use to test whether a similar interaction between endogenous and exogenous herpesvirus exists.

4 - Page 9, line 9: there seems to be a word missing in this sentence.

5 - Page 10, line 1: amniotes instead of amnions?

6 - Page 13, line 5: flank instead of flanks

7 - Page 18, line 16: it is unclear to me what the authors mean here by "reproduction"?

8 - Page 22, line 5: geological instead of geographical?

Reviewer #2 (Remarks to the Author):

This is a very interesting, and provocative manuscript describing a novel transposon subfamily in medaka fish. The peculiar feature of this transposon is that it apparently evolved out of a piggyBac element that received an ancient insertion of a herpesvirus.

I have the following observations:

1. The requirement for the internal TIR for transposition may suggest that the cleavage reactions by the transposase might actually occur at the internal TIR. Supplementary Fig. 8

shows the 5' end of the integrated transposon, but does that allow identification of which 5' TIR was actually used? In other words, transposition could take place at the external 5' TIR or at the internal 5' TIR. If transposition takes place at the internal TIR, then the entire plasmid is expected to integrate. This could be investigated by using Southern and a plasmid backbone specific probe, and/or by PCR, and/or by direct sequence analysis of integrants.

2. The most intriguing aspect of this study is unfortunately the least documented. Is the herpesvirus still "alive"? As the manuscript stands, the reader, although intrigued, is left with mixed feelings. It could well be, as authors describe in Discussion, that genomic integration of this herpesvirus was helpful in the distant past to generate tolerance against new herpesvirus infections. In the absence of further biological experiments, it could also be that the herpesvirus integration in this piggyBac transposon was a chance event that never had biological significance, and in that sense this herpesvirus genome is a mere passenger of piggyBac transposition.

Reviewer #3 (Remarks to the Author):

Very interesting paper that reports the largest eukaryotic transposons so far discovered. The fusion of a PiggyBac transposon with a complete Alloherpesvirus genome to form these Teratorn transposons certainly is a novel, remarkable phenomenon. Although preliminary results have been reported previously (Ref. 12), this paper presents definitive data on genome structure and transposase activity of Teratorn.

The main question that remains unanswered is: can a herpesvirus be activated from a Teratorn element? An experimental demonstration of such activation could be difficult to achieve and is probably unreasonable to expect as part of this paper. However, there are two amendments related to this point that the authors could and I think should implement to improve the paper. First, the sequences of the capsid proteins and proteins involved in morphogenesis (terminase, protease) should be analyzed in greater detail, to determine whether their sequences are compatible with virion formation (conservation of structural elements in the capsid proteins and the catalytic sites in the enzymes). Second, although the authors discuss the analogy with polintons, they seem to miss the key point, namely that most of the polintons encode two

capsid proteins along with the ATPase and protease required for virus formation, even though virions so far have not been discovered experimentally. Thus, the analogy between the polintons and Teratorn is actually quite complete and I think should be discussed along these lines. Furthermore, following the same theme, the data on virophage integration probably should be cited: Blanc G, Gallot-Lavallée L, Maumus F. Provirophages in the Bigelowiella genome

bear testimony to past encounters with giant viruses. Proc Natl Acad Sci U S A. 2015 Sep 22;112(38):E5318-26; Fischer MG, Hackl T. Host genome integration and giant virus-induced reactivation of the virophage mavirus. Nature. 2016 Dec 7;540(7632):288-291. Finally, the authors repeatedly state that most transposons only contain 1-3 genes which is somewhat disingenuous given that the widespread polintons are much larger.

Minor issues

The authors habitually use 'sequence homology', a common but wrong terminology. Should be 'sequence similarity' (observation) and 'homology' (conclusion).

The 'helicase' that is repeatedly mentioned in the text and Figure 2 - which one is this? UL9 homolog?

Line 4, p. 39:

"Only the 3rd codon was taken into account for the construction of phylogenetic trees." What is this supposed to mean: only the 3rd codon positions?

Figure 2a: amniotes not amnions

Ref 9: published in 2015 not 2014

Replies to the Reviewer 1 Comments:

We thank all Reviewers for giving us valuable comments and suggestions. We were pleased to know that all reviewers found our papers interesting. We have revised the manuscript following their comments and suggestions. Major revised portions are underlined in the text.

1-1) based on the presence of *Teratorn* in all four populations of medaka, the authors conclude that the element was present in the ancestor of these populations and that it has been transmitted vertically over 25 million years. In my opinion, this conclusion is weak because it is only based on indirect evidence. The congruence between the phylogeny of *Teratorn* and that of the medaka gene could be due to the fact that *Teratorn* is transmitted horizontally (in addition to vertically) on a regular basis, and that such transfers tend to occur between medaka individuals that are closely related both phylogenetically and geographically. Such horizontal transmission is plausible given that genes of viral origin that are involved in viral particle formation and virus replication are intact (devoid of non-sense mutations). *Teratorn* may well be inherited both vertically and horizontally and its presence in the medaka species may be much more recent than 25 myrs.

A direct evidence demonstrating the proposed antiquity of *Teratorn* would be to identify orthologous *Teratorn* copies between the various medaka strains, i.e., copies that are present at the same locus in two, three or even four strains. More specifically, a definitive conclusion on the presence of *Teratorn* in the medaka genome for 25 myrs requires the characterization of at least one copy shared at an orthologous locus between the HSQK or Nilan populations and at least one of the three other populations. Such a search for orthologous insertions requires a better characterization and comparison of the regions flanking each *Teratorn* copy in the four strains.

> Following the comment, we compared integration sites of *Teratorn* between the three medaka inbred strains (Hd-rR, HNI, HSOK), using the recently published genome data (http://utgenome.org/medaka_v2/#!Top.md). However, we failed to identify any orthologous copy in any pairs of the three strains (see Fig. 1 at pages

14–16 of this file). Thus, as the reviewer 1 pointed out, the possibility of horizontal transfer of *Teratorn* for each strain cannot be ruled out. I describe this possibility in the revised text (p.18. line 7~).

1-2) Based on the presence of a *Teratorn* element showing the same gene order in both the javanicus and latipes species group of the genus *Oryzias*, the authors conclude that this element originated in the common ancestor of these species and that it was inherited vertically during 60 Myrs. It has been shown that once integrated in a host genome, copies of mobile elements transmitted vertically in host populations typically evolve neutrally (e.g. Lampe et al. 2003 MBE). If *Teratorn* is present in *Oryzias* spp since more than 60 Myrs, many old and highly degraded copies should be found in the genome of these species and the average pairwise distance between copies within a given genome should be rather high, contrary to what is observed (Page 9, Line 3). Furthermore, if *Teratorn* is old, as proposed by the authors, some copies should be shared at orthologous locus between species from the latipes and javanicus groups. I thus reiterate my request of performing a search for such orthologous copies. Without such direct evidence for vertical inheritance, the authors cannot conclude without doubt that the element was present in the ancestor of the *Oryzias* genus 60 Myrs ago. Much like in the first paper, this point should be further explored. How many stop codons are expected during 60 Myrs under neutral evolution in teleosts? Are copies of *Teratorn* showing this expected number of stop codons observed? Would an alternative hypothesis that would imply a more recent age for *Teratorn* and involve a mix of vertical and horizontal transfer not be more plausible?

> We appreciate the previous and this comments very much to seriously consider the origin of *Teratorn*. This comment was given to the result of the second paper which is now included in the revised paper.

Given the absence of orthologous copies between the different strains of *Oryzias latipes* (Hd-rR, HNI, HSOK) as described above, we reasoned that there are no orthologous copy between the species of *latipes* and *javanicus* species group, too. In addition, we estimated the number of stop codons inside the two *Teratorn* genes (DNA polymerase and major capsid protein) under the assumption of neutral evolution, by simulating

neutral evolution from 1) the common ancestor of *O. latipes* and *O. dancena* to *O. latipes*, and 2) from the common ancestor of *O. latipes* and *O. mekongensis* to *O. latipes*. We found that the number of estimated stop codons was significantly larger than zero (no stop codon in current copies) (see Fig. 2 in page 17 of this file), arguing against the neutral evolution of *Teratorn* genes. Thus, as the reviewer 1 claimed, the possibility of horizontal transfer cannot be ruled out. Thus, in the revised version, we simply interpret the presence of *Teratorn* in the *Oryzias* genus as successful colonization of *Teratorn* in the host genomes of one genus. We also omit the description on the date (60 MYA) of integration timing of *Teratorn*, and mention that vertical and/or horizontal transfer are plausible (p.18. line 7~).

2) *Teratorn*, like other known genetic elements such as retroviruses and polintons, is at the boundary between viruses and transposable elements. Its transposase and terminal inverted repeats allow it to integrate and multiple itself into host genomes and its genes of viral origin may allow him to form infectious viral particles. In the discussion, the authors first conclude that *Teratorn* is a DNA transposon (p19, line 5) and then propose that it is a novel herpesvirus endogenized in the medaka genome (p21, line 7). This is confusing and may really impede the understanding of readers who are not familiar with the complexity of mobile genetic elements. The potentially dual nature of *Teratorn* should be more explicitly described and the authors should not attempt to provide a definitive conclusion on whether it is a virus or a transposon. They should also clearly mention that to be confirmed, the putative viral functions of *Teratorn* await observation of viral particles.

> We agree with this comment. We revised our entire manuscript not to definitively conclude that *Teratorn* is a virus or a transposon. In addition, we added the sentences describing the analogy between *Teratorn* and Polintons in the discussion part (p.26. line 4~) to emphasize the dual nature of *Teratorn*, a virus and a transposon.

3) The capacity of *Teratorn* to reactivate and form viral particles is tested by the authors

through treatment of medaka embryonic cells with chemicals that are known to reactivate latent human herpesviruses. No significant reactivation is observed. In a recent study, Fischer & Hackl (2016) show that the endogenous virophage Mavirus is reactivated by the presence of the Cafeteria roenbergensis virus, with which it shares promoter motifs. I was wondering whether the authors could have access to a strain of alloherpesvirus that they could use to test whether a similar interaction between endogenous and exogenous herpesvirus exists.

> We are really interested in the possibility of virus particle formation from *Teratorn* under the superinfection of related alloherpesvirus species. Unfortunately however, we cannot experimentally test this possibility at the moment by the following reasons. First, no exogenous herpesvirus has been identified so far which infects medaka under natural conditions. In addition, Yuan Y. et al. reported the lack of infectious capacity of Cyprinid herpesvirus 3 into medaka haploid ES cells (for reference, see below); Cyprinid herpesvirus 3 is the most intensively studied alloherpesvirus species because of the importance for marine fishery industry. As suggested by Reviewer 3, instead of experimental approaches, we analyzed the sequences of capsid proteins and proteins involved in virion morphogenesis (capsid maturation protease and DNA packaging terminase) to test whether their sequences are compatible with virion formation. We found that catalytic residues of the virion morphogenesis enzymes are conserved in *Teratorn* (data are included in revised Fig. 4a and Supplementary Fig. 9). In addition, we found the clear sequence similarity of major capsid protein and subunit 2 capsid triplex protein between *Teratorn* and exogenous alloherpesvirus species (data are included in new Supplementary Fig. 10). Thus, although virus function of *Teratorn* awaits virion detection, *Teratorn* could possibly be a “bona-fide” virus. We thus added a new sentence to the revised manuscript, “further experimental efforts to detect virions will be necessary to understand the life cycle of *Teratorn* and the biological significance of the existence of *Teratorn* in the medaka genome (p.27. line 5~)”.

*Reference : Yuan Y, *et al*, Medaka haploid embryonic stem cells are

susceptible to Singapore grouper iridovirus as well as to other viruses of aquaculture fish species. *J. Gen. Virol.* **94**, 2352-2359 (2013)

Minor points:

4 - Page 9, line 9: there seems to be a word missing in this sentence.

5 - Page 10, line 1: amniotes instead of amnions?

6 - Page 13, line 5: flank instead of flanks

7 - Page 18, line 16: it is unclear to me what the authors mean here by "reproduction"?

8 - Page 22, line 5: geological instead of geographical?

>We have corrected these minor points in the revised manuscript.

Replies to the Reviewer 2 Comments:

We thank all Reviewers for giving us valuable comments and suggestions. We were pleased to know that all reviewers found our papers interesting. We have revised the manuscript following their comments and suggestions. Major revised portions are underlined in the text.

1) The requirement for the internal TIR for transposition may suggest that the cleavage reactions by the transposase might actually occur at the internal TIR. Supplementary Fig. 8 shows the 5' end of the integrated transposon, but does that allow identification of which 5' TIR was actually used? In other words, transposition could take place at the external 5' TIR or at the internal 5' TIR. If transposition takes place at the internal TIR, then the entire plasmid is expected to integrate. This could be investigated by using Southern and a plasmid backbone specific probe, and/or by PCR, and/or by direct

sequence analysis of integrants.

>We think that, in this assay, chromosomal integration took place mainly via internal TIRs. In Supplementary Fig.7c, specific bands of 2.3-kb region for subtype 1 and 5.0-kb region for subtype 2 were detected in all colonies. These bands were derived from a DNA fragment that covers the right TIR and plasmid backbone produced by *HindIII* cut (Supplementary Fig. 7b, double-headed arrows). This data indicates the chromosomal integration via internal TIRs. Furthermore, in reply to this comment, we conducted inverse PCR using primers that specifically amplify integration sites mediated by either external or internal TIRs. We again identified integrated copies mediated by internal TIRs, but failed to obtain the evidence that supports integration via external TIRs. Thus, it is highly likely that internal TIRs were mainly used in this integration assay. We have no idea of why internal TIRs were mainly used for integration. The high frequency of internal TIR-mediated transposition could be due to the artificial circular configuration of the indicator plasmid. In any case, the aim of this *in vitro* assay was to test the activity of transposase and so we did not change the text in the revised manuscript.

2) The most intriguing aspect of this study is unfortunately the least documented. Is the herpesvirus still "alive"? As the manuscript stands, the reader, although intrigued, is left with mixed feelings. It could well be, as authors describe in Discussion, that genomic integration of this herpesvirus was helpful in the distant past to generate tolerance against new herpesvirus infections. In the absence of further biological experiments, it could also be that the herpesvirus integration in this *piggyBac* transposon was a chance event that never had biological significance, and in that sense this herpesvirus genome is a mere passenger of *piggyBac* transposition.

> Regarding the activity of herpesvirus, we analyzed the sequences of capsid proteins and proteins involved in virion morphogenesis (capsid maturation protease and DNA packaging terminase) to test whether their sequences are compatible with virion formation, as suggested by Reviewer

3. We found that catalytic residues of the virion morphogenesis enzymes are conserved in *Teratorn* (data are included in revised Fig. 4a and Supplementary Fig. 9). In addition, we found the clear sequence similarity of major capsid protein and subunit 2 capsid triplex protein between *Teratorn* and exogenous alloherpesvirus species (data are included in new Supplementary Fig. 10). Thus, although the virus function of *Teratorn* awaits virion detection, *Teratorn* could possibly be a “bona-fide” virus. We thus added a new sentence to the revised manuscript, “further experimental efforts to detect virions will be necessary to understand the life cycle of *Teratorn* and the biological significance of the existence of *Teratorn* in the medaka genome (p.27. line 5~)”.

Regarding the biological significance, we don't have any evidence to support the idea that *Teratorn* has some biological benefits to host organisms (medaka). However, we think that the fusion event has some biological significance for *Teratorn* itself; i.e. the fusion event enabled *Teratorn* to undergo intragenomic propagation. We do not think that *Teratorn* was created by an integration of herpesvirus into a *piggyBac* transposon and that the virus is a mere passenger of *piggyBac* transposition. If *Teratorn* was created by a chance event of integration as the reviewer suggests, copies of the same *piggyBac* transposon should exist without herpesvirus sequences. However, as presented in Supplementary Fig. 5, there is no such copy in the medaka genome.

Replies to the Reviewer 3 Comments:

We thank all Reviewers for giving us valuable comments and suggestions. We were pleased to know that all reviewers found our papers interesting. We have revised the manuscript following their comments and suggestions. Major revised portions are underlined in the text.

1) The main question that remains unanswered is: can a herpesvirus be activated from a

Teratorn element? An experimental demonstration of such activation could be difficult to achieve and is probably unreasonable to expect as part of this paper. However, there are two amendments related to this point that the authors could and I think should implement to improve the paper.

1-1) First, the sequences of the capsid proteins and proteins involved in morphogenesis (terminase, protease) should be analyzed in greater detail, to determine whether their sequences are compatible with virion formation (conservation of structural elements in the capsid proteins and the catalytic sites in the enzymes).

> Following the suggestion, we analyzed the sequences of capsid proteins and proteins involved in virion morphogenesis (capsid maturation protease and DNA packaging terminase) to test whether their sequences are compatible with virion formation. We found that catalytic residues of the virion morphogenesis enzymes are conserved in *Teratorn* (data are included in revised Fig. 4a and Supplementary Fig. 9). In addition, we found the clear sequence similarity of major capsid protein and subunit 2 capsid triplex protein between *Teratorn* and exogenous alloherpesvirus species (data are included in new Supplementary Fig. 10). Thus, although virus function of *Teratorn* awaits virion detection, *Teratorn* could possibly be a “bona-fide” virus. We thus added a new sentence to the revised manuscript, “further experimental efforts to detect virions will be needed to understand the life cycle of *Teratorn* and the biological significance of the existence of *Teratorn* in the medaka genome (p.27. line 5~)”.

1-2) Second, although the authors discuss the analogy with polintons, they seem to miss the key point, namely that most of the polintons encode two capsid proteins along with the ATPase and protease required for virus formation, even though virions so far have not been discovered experimentally. Thus, the analogy between the polintons and *Teratorn* is actually quite complete and I think should be discussed along these lines.

> Following the suggestion, we added description on the analogy between the two mobile elements in the paragraph of the discussion part, pointing

out the potential to produce virus particles (p.26. line 4~).

1-3) Furthermore, following the same theme, the data on virophage integration probably should be cited: Blanc G, Gallot-Lavallée L, Maumus F. Provirophages in the *Bigelowiella* genome bear testimony to past encounters with giant viruses. *Proc Natl Acad Sci U S A*.2015 Sep 22;112(38):E5318-26; Fischer MG, Hackl T. Host genome integration and giant virus-induced reactivation of the virophage mavirus. *Nature*. 2016 Dec 7;540(7632):288-291.

> We agree that those two papers are very important and provides some implications for the remaining questions of how *Teratorn* is activated to produce virus particles and of why *Teratorn* exists in the medaka genome. Thus, we cited these two papers in the paragraph of the discussion part, in the light of its biological significance for their hosts (p.26. line 18~).

1-4) Finally, the authors repeatedly state that most transposons only contain 1-3 genes which is somewhat disingenuous given that the widespread polintons are much larger.

> Following this comment, we deleted this sentence from our manuscript.

Minor issues

2-1) The authors habitually use 'sequence homology', a common but wrong terminology. Should be 'sequence similarity' (observation) and 'homology' (conclusion).

> Following this comment we corrected the wording of the two terms.

2-2) The 'helicase' that is repeatedly mentioned in the text and Figure 2 - which one is this? UL9 homolog? Line 4, p. 39:

> As pointed out, the helicase gene in *Teratorn* is UL9 homolog. To avoid confusion, we modified our text : "...such as DNA replication (DNA polymerase, primase and UL21 homolog DNA helicase), ..." (p.9. line 1).

2-3) Figure 2a: amniotes not amnions

2-4) Ref 9: published in 2015 not 2014

>We have corrected these minor points in the revised manuscript.

Subtype 1 Left

Figure 1

* : Insert into the same repetitive sequence

** : No overlap except for TIR

*** : internal TIR

Subtype 1 right

Figure 1 (continued)

Subtype 2 Left

Subtype 2 Right

Figure 1 | No orthologous *Teratorn* copy among the three medaka inbred strains. Neighbor-joining trees based on the sequences of TIR and its flanking 100 bp region of all identified *Teratorn* copies of the three medaka inbred strains (Hd-rR, magenta; HNI, blue; HSOK, orange). These trees were constructed by pairwise-deletion method. Note that the flanking sequences derived from different strains aren't clustered together closely, except for the case in which *Teratorn* was inserted into the same repetitive region of each strain (* for subtype 1). These data indicate that there are no orthologous *Teratorn* copy among the three strains.

Figure 2 | Estimation of the number of stop codons of *Teratorn* genes under neutral mutation in the genus *Oryzias*. **a**) A maximum-likelihood tree of medaka related species based on the 3rd codon positions of nine nuclear genes (*glyt*, *sh3px3*, *rag1*, *ptch1*, *tbr*, *myh6*, *zic1*, *plagl2*). Since the divergence time and evolutionary rate is ambiguous in the genus *Oryzias*, we instead estimated evolutionary distances between those species from this tree, assuming that 3rd codon positions undergo neutral evolution. **b**) Maximum-likelihood trees of DNA polymerase and major capsid protein used for inference of ancestral sequences of each gene (dark blue, common ancestor of *O. latipes* and *O. dancena*; cyan, common ancestor of *O. latipes* and *O. mekongensis*). **c**) Histograms of the number of stop codons created in the two *Teratorn* genes under neutral evolution, from the common ancestor of *O. latipes* and *O. dancena* to *O. latipes* (upper) and from the common ancestor of *O. latipes* and *O. mekongensis* to *O. latipes* (lower). Ancestral sequences of the two *Teratorn* genes were evolved neutrally under HKY model for 100,000 times by using Seq-Gen Ver 1.3.3¹ as previously described². Note that the numbers of stop codons under neutral evolution are larger than those in the current *Teratorn* copies (zero), implying selection and / or recent invasion of *Teratorn* in the genus *Oryzias*.

1. Rambaut A, Grassly NC. Seq-Gen: an application for the Monte Carlo simulation of DNA sequence evolution along phylogenetic trees. *Comput Appl Biosci*. 13(3):235-8 (1997)
2. Kobayashi Y. et al, No evidence for Natural Selection on Endogenous Borna-like Nucleoprotein Elements after the Divergence of Old World and New World Monkeys. *Plos One*, 6(9):e24403 (2011)

REVIEWERS' COMMENTS:

Reviewer #1 (Remarks to the Author):

I am satisfied by the way the authors have addressed my comments and those of the other reviewers. I have no additional comment to make.

Reviewer #2 (Remarks to the Author):

1) Concerning my question, whether transposition occurs at the external TIRs or at the internal TIRs, I am not completely confused. The excision experiment shown in Fig. 3a,b clearly indicates catalytic transposase activity at the external TIRs. Yet, in their rebuttal letter authors claim that "We think that, in this assay, chromosomal integration took place mainly via internal TIRs." They also state: "Furthermore, in reply to this comment, we conducted inverse PCR using primers that specifically amplify integration sites mediated by either external or internal TIRs. We again identified integrated copies mediated by internal TIRs, but failed to obtain the evidence that supports integration via external TIRs. Thus, it is highly likely that internal TIRs were mainly used in this integration assay." These statements are in clear contradiction with data presented in Fig. 3. Please clarify.

2) In the rebuttal letter, in response to my question authors argue: "We do not think that Teratorn was created by an integration of herpesvirus into a piggyBac transposon and that the virus is a mere passenger of piggyBac transposition. If Teratorn was created by a chance event of integration as the reviewer suggests, copies of the same piggyBac transposon should exist without herpesvirus sequences. However, as presented in Supplementary Fig. 5, there is no such copy in the medaka genome." Just because the medaka genome does not contain "empty" piggyBac transposons does not rule out that those elements exist somewhere else. Authors now discuss a scenario, in which horizontal gene transfer played a role in distributing Teratorn elements in medaka species. Thus, it may well be that the fusion of a piggyBac transposon and the herpesvirus genome took place in another, unknown genome and the resulting element has been horizontally transferred to medaka, followed by several

rounds of transposition thereby generating extra copies. I believe this is a plausible scenario. Also, for the sake of argumentation, what was the acquisition of the herpesvirus genome by the piggyBac transposon if not a chance event? Please discuss this in a concise and clear manner.

3) Authors state in Discussion: "For example, several studies reported the insertion of an insect transposon into a baculovirus genome. Indeed, all viruses have the potential to shift into the intragenomic life cycle, if they acquire an integration system from other sources." This is a confusing argument. If a transposon integrates into a virus, then the transposon might gain the ability to spread within and between species through the infectious potential of the virus. And just the other way around, if a virus integrates into a transposon (like in Teratorns), then the virus might become endogenized. This has to be clearly discussed.

Reviewer #3 (Remarks to the Author):

My comments as well as (to the best of my judgment) those of the other reviewers have been addressed thoroughly. In particular, by combining the material from the two originally submitted manuscripts into this single article, the authors succeeded in producing a compelling paper. I have no further critical comments.

Replies to the Reviewer 2 Comments:

We thank Reviewer 2 for giving us valuable comments and suggestions. We have revised the manuscript following the comments and suggestions. Major revised portions are underlined in the text.

1) Concerning my question, whether transposition occurs at the external TIRs or at the internal TIRs, I am not completely confused. The excision experiment shown in Fig. 3a,b clearly indicates catalytic transposase activity at the external TIRs. Yet, in their rebuttal letter authors claim that "We think that, in this assay, chromosomal integration took place mainly via internal TIRs." They also state: "Furthermore, in reply to this comment, we conducted inverse PCR using primers that specifically amplify integration sites mediated by either external or internal TIRs. We again identified integrated copies mediated by internal TIRs, but failed to obtain the evidence that supports integration via external TIRs. Thus, it is highly likely that internal TIRs were mainly used in this integration assay." These statements are in clear contradiction with data presented in Fig. 3. Please clarify.

> We think that excision reaction occurs both via a pair of external TIRs and internal TIRs in this assay, although the latter case is difficult to test by PCR since the length of the remaining DNA is only about 50 bp. However, chromosomal integration was less efficient at external TIRs than at internal TIRs, based on our data. At the moment, we do not know the reason for the lower efficiency of external TIRs for integration. This happened only *in vitro* culture cells but might not *in vivo*, since there is no *Teratorn* copy that was integrated via internal TIRs in the medaka genome. Anyway, we revised our manuscript as clearly as possible to explain the relationship between the presence / absence of internal TIRs and the result of the excision / integration assay (p.12. line 221~).

2) In the rebuttal letter, in response to my question authors argue: "We do not think that *Teratorn* was created by an integration of herpesvirus into a piggyBac transposon and that the virus is a mere passenger of piggyBac transposition. If *Teratorn* was created by a chance event of integration as the reviewer suggests, copies of the same piggyBac transposon should exist without herpesvirus sequences. However, as presented in Supplementary Fig. 5, there is no such copy in the medaka genome." Just because the medaka genome does not contain "empty" piggyBac transposons does not rule out that those elements exist somewhere else. Authors now discuss a scenario, in which horizontal gene transfer played a role in distributing *Teratorn* elements in medaka species. Thus, it may well be that the fusion of a piggyBac transposon and the herpesvirus genome took place in another, unknown genome and the resulting element has been horizontally transferred to medaka, followed by several rounds of transposition thereby generating extra copies. I believe this is a plausible scenario. Also, for the sake of argumentation, what was the acquisition of the herpesvirus genome by the piggyBac transposon if not a chance event? Please discuss this in a concise and clear manner.

> Not clearly stated in the text, we agree that the first fusion event happened by chance, probably in somewhere else other than medaka, and that the invasion of medaka *Teratorn* was the result of horizontal transfer. What happened at the very early stage of the formation of *Teratorn* is largely unknown. We think that there are at least two possibilities; one is the integration of the herpesvirus genome into a *piggyBac*, as pointed out by Reviewer 2, while the other possibility is the integration of the *piggyBac* transposon into a latently infected herpesvirus genome floating in the nucleus. We added the above scenario in 'Discussion' (p.24. line 436~).

3) Authors state in Discussion: "For example, several studies reported the insertion of an insect transposon into a baculovirus genome. Indeed, all viruses have the potential to shift into the intragenomic life cycle, if they acquire an

integration system from other sources." This is a confusing argument. If a transposon integrates into a virus, then the transposon might gain the ability to spread within and between species through the infectious potential of the virus. And just the other way around, if a virus integrates into a transposon (like in Teratorns), then the virus might become endogenized. This has to be clearly discussed.

> We agree with this comment. The sentence "For example, several studies reported the insertion of an insect transposon into a baculovirus genome" is confusing and not appropriate here, because this example only tells the case that transposons jumped into virus genomes. Our speculation is that the fusion event took place by integration of the *piggyBac* transposon into a latently infected herpesvirus genome in the nucleus, but the possibility that the integration of the herpesvirus genome into a *piggyBac* can not be ruled out, as described above. We therefore deleted this sentence from 'Discussion' (p.28. line 500). This does not affect our points in 'Discussion'.

REVIEWERS' COMMENTS:

Reviewer #2 (Remarks to the Author):

Authors have addressed my concerns in a satisfactory manner.